# PSEUDO-LABELS ARE ALL YOU NEED FOR OUT-OF-DISTRIBUTION DETECTION

## ABSTRACT

Detecting out-of-distribution (OOD) samples is a significant challenge in real-world deep-learning applications, such as medical imaging and autonomous driving. Traditional machine learning models, primarily trained on in-distribution (ID) data, often struggle when encountering OOD instances, resulting in unreliable predictions. While supervised OOD detection methods generally outperform unsupervised approaches due to the availability of labeled data, our research uncovers a crucial insight: their success is not necessarily due to recognizing the actual object categories in the images; instead, these methods rely on a specific classification strategy that may not correspond to real-world understanding. Essentially, supervised methods detect OOD samples by identifying the difficulties in classifying unfamiliar data. This challenge is similar to what unsupervised OOD detection methods face, as they also depend on the failure to reconstruct OOD data due to the lack of prior exposure. In this study, we bridge the gap between supervised and unsupervised OOD detection by introducing a novel approach that trains models to classify data into pseudo-categories. We employ self-supervised learning (SSL) to convert raw data into representations, which are then clustered to generate pseudo-labels. These pseudo-labels are subsequently used to train a classifier, enabling its OOD detection capabilities. Experimental results show that our approach surpasses state-of-the-art techniques. Furthermore, by training models on different sets of pseudo-labels derived from the dataset, we enhance the robustness and reliability of our OOD detection method.

## 1 INTRODUCTION

In real-world applications, machine learning models often encounter out-of-distribution (OOD) data instances that differ from the classes seen during training. While models may perform well on in-distribution (ID) data, detecting and managing OOD data is crucial for ensuring model trustworthiness. Robust OOD detection mechanisms not only prevent erroneous predictions in unfamiliar scenarios but also enhance model reliability, robustness, and safety across diverse application contexts.

Current OOD detection methods can be roughly categorized into supervised and unsupervised approaches. Supervised methods rely on labeled datasets to learn the characteristics of ID data and distinguish them from OOD samples. These methods typically learn to classify image labels and use this learned knowledge to identify deviations from the learned data distribution as potential OOD samples. These methods are known for their speed and generally high accuracy, but their reliance on extensive human-labeled data can be a significant drawback, especially in domains where labeling is costly or impractical. In contrast, unsupervised methods do not require labeled data and focus on reconstructing images using powerful generative models such as autoencoders, generative adversarial networks, and diffusion models. While these unsupervised approaches can be slow due to the complexity of the models, they are advantageous in scenarios where labeled data is scarce. Despite many studies exploring both supervised and unsupervised methods, their detection accuracies are not directly comparable due to different experimental settings.

We first compared state-of-the-art (SOTA) supervised and unsupervised methods in two distinct experimental settings, with the results shown in Table 1. One setting is typically used to evaluate supervised methods, while the other is common for unsupervised methods. It is evident that the supervised SOTA, FeatureNorm (Yu et al., 2023), outperformed the unsupervised SOTA, DDPM

| Method | Unsupervised Benchmark Setting | Supervised Benchmark Setting | | Backbone | Time (s) |
|---|---|---|---|---|---|
| | AUROC ↑ | AUROC ↑ | FPR95 ↓ | | |
| DDPM (Graham et al., 2023) | 79.16[*] | 94.21 | 27.60 | U-Net | $28.32 \pm 1.5$ |
| FeatureNorm (Yu et al., 2023) | 81.76 | 97.33[*] | 13.53[*] | WRN-28-10 | $0.025 \pm 0.007$ |

Table 1: We compared state-of-the-art supervised and unsupervised methods using their respective benchmark settings. The supervised method is FeatureNorm, based on the WRN-28-10 (Zagoruyko & Komodakis, 2016) backbone, while the unsupervised method is DDPM, utilizing a U-Net (Ronneberger et al., 2015) backbone. The table presents the average area under the receiver operating characteristic curve (AUROC) and the false positive rate at 95% true positive rate (FPR95) (Liang et al., 2017), where a higher AUROC and lower FPR95 indicate better performance. Values marked with an asterisk (*) are taken directly from the referenced papers. Timings were measured on an NVIDIA RTX A4000 GPU, processing a batch of 16 images.

(Graham et al., 2023), in both efficiency and accuracy. This performance advantage is likely due to the semantic information provided by human labels, which allows models to learn more discriminative features. However, we hypothesize that the success of supervised methods in detecting OOD data is not solely driven by human knowledge. Instead, it may be their ability to learn structured classification strategies that leads to their effectiveness. In this study, we explore whether pseudo-labels, which are generated automatically rather than manually, can also be effective for OOD detection. To generate pseudo-labels, we experiment with three different strategies: (1) random assignment, (2) clustering based on raw data, and (3) clustering based on self-supervised learning (SSL) representations.

The results demonstrate that our proposed method, which combines pseudo-labeling with SSL representations and clustering, outperforms both current supervised and unsupervised methods. Through comprehensive experiments, we achieve a balance between accuracy and efficiency without the need for extensive labeled data. We also explore ensembling different sets of pseudo-labels derived from the ID dataset, which further enhances the performance of our model. This ensemble approach leverages the diversity of pseudo-labels to capture a broader range of underlying patterns, resulting in more robust and accurate OOD detection performance.

## 2 BACKGROUND

**Supervised Methods.** Supervised OOD detection methods depend on learning from labeled data to distinguish between in-distribution (ID) and out-of-distribution (OOD) samples. Hendrycks & Gimpel (2016) introduced a baseline approach that utilized softmax confidence scores for OOD detection, which was later enhanced by Liang et al. (2017) through the use of temperature scaling and input perturbations. Lee et al. (2018) further advanced the field by incorporating Mahalanobis distances between test samples and the ID dataset. However, many of these methods suffer from overconfidence when encountering OOD samples. To address this issue, DeVries & Taylor (2018) proposed a strategy where models are incentivized to make accurate predictions by optimizing both prediction loss and an additional confidence loss, ensuring that high confidence scores more accurately reflect the model's prediction capability. Liu et al. (2020) tackled the overconfidence problem by measuring how well an input fits within the learned data distribution. More recently, innovations such as contrastive training for robust feature learning (Tack et al., 2020), leveraging large-scale pre-trained models (Fort et al., 2021), and post-hoc scoring methods like LogitNorm (Wei et al., 2022), MaxLogit, and MaxCosine (Zhang & Xiang, 2023) have significantly improved OOD detection. Additionally, new insights have emerged from using feature map norms from intermediate network layers, as explored by Yu et al. (2023). These developments highlight the ongoing advancements aimed at enhancing the reliability and robustness of OOD detection in supervised settings.

**Unsupervised Methods.** To reduce reliance on labeled data, researchers have turned to unsupervised OOD detection methods. These approaches leverage techniques such as reconstruction error and density estimation, eliminating the need for human-annotated labels. Autoencoders were among the earliest methods employed, demonstrating their effectiveness in anomaly detection based on reconstruction error (An & Cho, 2015). Generative adversarial networks (GANs) further advanced the field through adversarial training (Schlegl et al., 2017). In addition to reconstruction methods,

density estimation techniques have also shown promising results. For example, Ren et al. (2019) proposed Likelihood Ratios, which assess the likelihood of a sample under an ID model compared to that under a broader generative model. However, it has been observed that relying solely on likelihood scores from deep generative models can be problematic, as these scores do not always serve as a reliable metric for OOD detection (Nalisnick et al., 2018; Kamkari et al., 2024). More recent methods have explored alternative approaches using normalizing flows (Cook et al., 2024) and transformer-based models (Podolskiy et al., 2021), which offer improved modeling of complex data distributions and provide high-quality reconstructions for OOD detection. Moreover, diffusion-based methods have recently shown great promise. These techniques measure discrepancies between input images and their reconstructed counterparts to detect OOD samples (Graham et al., 2023; Liu et al., 2023). Despite these advancements, unsupervised methods tend to be slower and still lag behind supervised methods in terms of performance when directly compared.

**Pseudo-labels.** Pseudo-labeling has been effectively applied across various fields, such as semi-supervised learning (Lee et al., 2013), domain adaptation (Litrico et al., 2023), and unsupervised learning (Caron et al., 2018), by utilizing unlabeled data to enhance model performance. The integration of clustering techniques with self-supervised learning approaches (Caron et al., 2018; Chen et al., 2020; Van Gansbeke et al., 2020) has shown remarkable success in generating reliable pseudo-labels, thereby reducing the need for large, annotated datasets. Building on these advancements, we explore the use of pseudo-labeling as a substitute for human-annotated labels in OOD detection.

## 3 METHODOLOGY

### 3.1 ANALYSIS OF CURRENT SUPERVISED AND UNSUPERVISED OUT-OF-DISTRIBUTION DETECTION METHODS

Supervised and unsupervised OOD detection methods are built on distinct principles. Supervised approaches focus on classifying samples, identifying them as OOD when their predicted labels are less confident. In contrast, unsupervised methods consider samples as OOD if their representations fall in low-density regions or exhibit high reconstruction errors. However, these methods are typically evaluated under different experimental conditions. Comparing them under consistent settings allows for a clearer understanding of their relative strengths and the actual impact of using labels. As illustrated in Table 1, supervised methods outperform unsupervised ones in both speed and accuracy of OOD detection. Mapping image representations to specific labels proves effective in distinguishing between ID and OOD data. However, since supervised methods heavily depend on labor-intensive ground-truth (GT) labels, our goal is to assess whether replacing GT labels with pseudo-labels can maintain comparable OOD detection performance.

### 3.2 HYPOTHESIS: GROUND-TRUTH LABELS ARE NOT NECESSARY

We hypothesize that the model's ability to detect OOD samples may not stem from recognizing the true object categories but rather from learning *a specific classification strategy that doesn't necessarily align with real-world understanding*. In essence, while these methods excel in classifying ID data, they often struggle with OOD samples, which they have never encountered. This challenge mirrors the limitations seen in unsupervised methods, which also fail to reconstruct OOD data effectively due to their unfamiliarity.

**Pilot Study.** To investigate the viability of using pseudo-labels as a substitute for GT labels for OOD detection, we conducted a pilot study where models were trained using three different types of labels. In this study, we used the MNIST dataset (LeCun et al., 1998) as the ID data and the FashionMNIST dataset (Xiao et al., 2017) as the OOD data. The MNIST dataset was labeled in three distinct ways: with ground-truth labels, randomly assigned labels, and cluster-based labels obtained from k-means applied in the image space. Each labeling method resulted in 10 categories. We trained a ResNet18 model (He et al., 2016) to classify the ID data. After training, we computed the FeatureNorm (Yu et al., 2023) for each sample to generate OOD scores, following the methodology used in prior studies. The model's OOD detection performance was evaluated using a test set that combined the MNIST test dataset and the FashionMNIST dataset. Additionally, the MNIST training set was split into two parts: 95% for training and 5% for evaluating classification accuracy.

| Method | FPR95 ↓ | AUROC ↑ | Test Accuracy (%) |
|---|---|---|---|
| Random Assignment | 46.56 | 84.25 | 10.72 |
| Raw image clustering | 31.87 | 93.22 | 94.17 |
| GT labels | 10.18 | 97.38 | 99.49 |

Table 2: FPR95 and AUROC metrics were used to evaluate the OOD detection performance of models trained with data labels generated using various strategies. The results demonstrate a positive correlation between OOD detection performance and classification accuracy on the test set.

**Findings and Analysis.** Table 2 summarizes the results of this pilot study. The model trained with GT labels achieved the best performance, exhibiting the lowest false positive rate at 95% true positive rate (FPR95) (Liang et al., 2017) and the highest area under the receiver operating characteristic curve (AUROC). The model trained with cluster-based labels from raw image clustering performed moderately, while the model trained with random labels showed the poorest performance. Notably, the models' OOD detection capabilities were consistent with their classification accuracy on the testing data: higher testing accuracy corresponded to better OOD detection performance.

These findings highlight the critical role of label quality in training effective classification models for OOD detection. The model trained with random labels failed to learn meaningful classification rules and could not effectively distinguish between ID and OOD samples. The model trained with cluster-based labels from raw images performed better than random labeling but still fell short compared to the model trained with GT labels. This is likely because clustering in the raw image space captures low-level features like pixel intensities and colors, which do not align well with the high-level semantic features that modern classification networks extract.

The moderate performance of the raw image clustering model underscores the need for pseudo-labels derived from high-level representations. Since classification networks like ResNet18 reduce spatial resolution and focus on abstract features, labels based on high-level semantic information are more conducive to learning effective classification strategies. SSL methods are particularly well-suited for this purpose, as they can learn rich, high-level features without requiring labeled data.

By deriving pseudo-labels from high-level representations obtained through SSL, we can capture the semantic content of the images more effectively. This approach is expected to improve both classification accuracy and OOD detection performance, bringing models trained with pseudo-labels closer to those trained with GT labels. Thus, the pilot study demonstrates the importance of using SSL pseudo-labels to enhance OOD detection capabilities.

## 3.3 PSEUDO-LABEL GENERATION USING SELF-SUPERVISED METHODS

We utilize the MBT method (Gedara Chaminda Bandara et al., 2023), which builds on the Barlow Twins (Zbontar et al., 2021) architecture – a popular Siamese network – to process distinct views of a batch of images $X$, denoted as $Y^A$ and $Y^B$. These views are created by applying random augmentations $\mathcal{T}$ to $X$. As illustrated in Figure 1 (a), $Y^A$ and $Y^B$ pass through an encoder $f_e$ and a projector $f_p$ to generate normalized embeddings $Z^A$ and $Z^B$. Additionally, a regularization branch processes interpolated images $Y^M = \lambda Y^A + (1 - \lambda)Y^{B'}$, where $Y^{B'}$ represents a shuffled batch from $Y^B$. The output of this regularization branch is denoted as $Z^M$. These embeddings are then used to compute the Barlow Twins loss $\mathcal{L}_{BT}$ and Mixup (Zhang et al., 2017) regularization objective $\mathcal{L}_{reg}$. The final loss function is:

$$\mathcal{L} = \mathcal{L}_{BT} + \lambda_{\text{reg}}\mathcal{L}_{\text{reg}}. \tag{1}$$

In the final loss, $\mathcal{L}_{BT}$ encourages similar representations for different augmentations of the same image, formulated as:

$$\mathcal{L}_{BT} = \sum_i \left(1 - \frac{\langle z^A_{.,i}, z^B_{.,i}\rangle_b}{\left\|z^A_{.,i}\right\|_2 \left\|z^B_{.,i}\right\|_2}\right)^2 + \lambda_{BT}\sum_i\sum_{j \neq i}\left(\frac{\langle z^A_{.,i}, z^B_{.,j}\rangle_b}{\left\|z^A_{.,i}\right\|_2 \left\|z^B_{.,j}\right\|_2}\right)^2 \tag{2}$$

where $b$ indexes batch samples and $i, j$ index feature dimensions. $\lambda_{BT}$ is a weighting factor. The dot in $z^A_{.,i}$ implies that all features from the batch are considered. The first term of the Barlow Twins loss

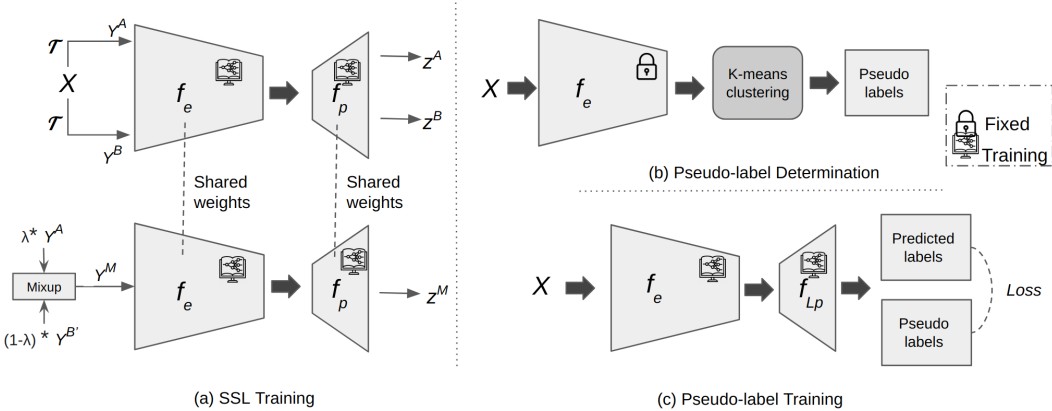

Figure 1: An overview of our pipeline: (a) We first train a network using self-supervised learning to extract relevant features from the in-distribution (ID) dataset. The dotted vertical lines signify that the same network is used, employing shared weights. (b) Next, we apply K-means clustering to group the extracted features and generate pseudo-labels. (c) Finally, we train the network to classify images into pseudo-categories and compute feature norms to detect OOD samples.

minimizes the variance of feature dimensions when aligning representations of different views of the same image, while the second term reduces redundancy between the dimensions.

The regularization term $\mathcal{L}_{reg}$ in final loss enhances Barlow Twins training by encouraging linearly interpolated inputs to produce linearly interpolated features. This adds regularization to the original objective. Let $Z^{M'} = \lambda Z^A + (1 - \lambda)Z^{B'}$ represent features interpolated in the embedding space. The mixup regularization $\mathcal{L}_{reg}$ is defined as:

$$\mathcal{L}_{\text{reg}} = \left\| (Z^M)^\top Z^A - (Z^{M'})^\top Z^A \right\|_2 + \left\| (Z^M)^\top Z^B - (Z^{M'})^\top Z^B \right\|_2. \tag{3}$$

Here, $Z^M$ refers to features derived from images interpolated in the image space, while $Z^{M'}$ refers to features interpolated in the embedding space. After SSL training, the generated features are clustered using K-means (MacQueen et al., 1967), and the resulting clusters provide pseudo-labels, which are essential for the OOD training phase.

## 3.4 OUT-OF-DISTRIBUTION DATA DETECTION BASED ON PSEUDO-LABEL TRAINING

In this step, we trained a classifier (see Figure 1 (c)) to categorize images using the pseudo-labels generated from the clustering process. The underlying principle is similar to supervised OOD detection methods, where the model learns the internal structure of an ID dataset. Once the network is trained, we calculate the feature map norm of the convolution block (Yu et al., 2023) to classify samples as either ID or OOD. Specifically, given a feature map $z \in \mathbb{R}^{C \times W \times H}$ obtained from the trained classifier $f_e$, the average norm across all channels in a block $B$ is computed as:

$$f_{\text{FeatureNorm}}(x; B) = \frac{1}{C} \sum_{c=1}^{C} \sqrt{\sum_{w=1}^{W} \sum_{h=1}^{H} \max(z_c(w, h), 0)^2}, \tag{4}$$

where $z_c \in \mathbb{R}^{1 \times W \times H}$ represents the feature map of the $c$-th channel, and $\max(z_c(w, h), 0)$ rectifies values by ignoring negative elements. This norm reflects the average activation level of the feature map for the data $X$ in the block $B$. Following (Yu et al., 2023), samples with high norm values are considered ID, while those with low norm values are classified as OOD (Park et al., 2023). Additionally, we use the convolution block with the highest norm ratios – determined based on the response to ID data and their augmented variants – to distinguish ID from OOD samples.

| | FMnist | | | CIFAR-10 | | | | CelebA | | | | SVHN | | | | Avg. Rank |
|---|---|---|---|---|---|---|---|---|---|---|---|---|---|---|---|---|
| | Mnist | V | H | Svhn | CelA | V | H | Cif10 | Svhn | V | H | Cif10 | CelA | V | H | |
| Likelihood(Bishop, 1994) | 8.5 | 55.5 | 51.1 | 6.1 | 52.3 | 50.5 | 50.0 | 67.4 | 5.9 | 57.9 | 50.1 | 99.0 | 99.9 | 50.2 | 50.3 | 5.7 |
| WAIC(Choi et al., 2018) | 8.8 | 55.4 | 51.1 | 6.0 | 52.5 | 50.4 | 50.0 | 67.0 | 5.7 | 57.8 | 50.1 | 99.0 | 99.9 | 50.2 | 50.3 | 6.1 |
| Typicality(Nalisnick et al., 2018) | 81.1 | 51.2 | 49.6 | 88.6 | 54.0 | 51.2 | 50.0 | 66.5 | 88.5 | 51.0 | 50.0 | 97.1 | 98.9 | 50.0 | 49.9 | 6.3 |
| DOS(Morningstar et al., 2021) | 98.1 | 67.1 | 55.3 | 96.4 | 54.6 | 51.0 | 50.1 | 84.9 | 99.4 | 69.6 | 49.9 | 98.9 | 99.9 | 50.3 | 50.9 | 3.8 |
| AutoEncoder | 75.0 | 59.0 | 50.4 | 3.2 | 75.3 | 50.1 | 49.9 | 42.1 | 2.7 | 53.2 | 49.9 | 99.4 | 99.9 | 49.9 | 49.9 | 6.5 |
| AE_MH(Denouden et al., 2018) | 94.9 | 79.5 | 63.0 | 4.5 | 71.8 | 50.9 | 50.0 | 64.2 | 9.6 | 71.6 | **50.8** | 99.8 | **100** | 50.6 | 50.5 | 3.9 |
| MemAE(Gong et al., 2019) | 56.9 | 59.0 | 48.7 | 4.2 | 69.4 | 50.4 | 49.9 | 51.5 | 5.8 | 56.4 | 49.9 | 98.6 | 99.5 | 49.9 | 49.9 | 6.8 |
| AnoDDPM-Mod(Wyatt et al., 2022) | 91.8 | 81.0 | 64.2 | 5.0 | 54.2 | 50.0 | 49.9 | 67.3 | 78.4 | 59.4 | 50.2 | 98.0 | 97.8 | 50.2 | 52.7 | 5.2 |
| DDPM(Graham et al., 2023) | **97.4** | **88.6** | 65.1 | 97.9 | 68.5 | **63.2** | **50.5** | 99.0 | **100** | 93.3 | 50.3 | 94.2 | 99.6 | 58.2 | **61.6** | 2.6 |
| NAE(Kamkari et al., 2024) | 95.1 | - | - | 93.6 | 65.5 | - | - | - | - | - | - | 98.7 | 99.6 | - | - | - |
| Our method | 96.7 | 87.0 | **70.5** | **99.1** | **76.9** | **70.2** | 49.9 | **99.8** | **100** | **98.5** | 50.7 | 99.5 | 99.0 | **67.1** | **66.5** | **1.8** |

Table 3: We compared our approach with unsupervised methods using their respective benchmark settings. In the table, the first and second rows represent the ID and OOD datasets, while "V" and "H" indicate vertically and horizontally flipped images of the ID dataset. OOD detection performance is evaluated using AUROC scores, with baseline method scores taken from (Graham et al., 2023) and (Kamkari et al., 2024). The best and second-best results are highlighted in bold and underlined, respectively.

## 3.5 Implementation Details

Our method involves two stages: SSL training and classifier training, both conducted on the ID dataset. For the SSL stage, we adhered to the default hyperparameter settings outlined in (Gedara Chaminda Bandara et al., 2023). In the classifier training stage, we initialized the network with the SSL-trained encoder $f_e$, and trained $f_e$ and $f_{Lp}$ using an SGD optimizer with a batch size of 128, momentum of 0.9, and a weight decay of 0.01. Given that our pilot study results indicate OOD detection performance is influenced by the classifier's generalization ability, we randomly selected 5% of the ID training set for accuracy evaluation and model selection. Following the approach in (Yu et al., 2023), we identified the block with the highest norm ratio to compute the OOD score.

## 4 Experiments and Evaluations

We assessed the OOD detection performance of our approach by comparing it against several supervised and unsupervised methods. The experiments were conducted using ResNet18 (He et al., 2016), VGG11 (Simonyan & Zisserman, 2014), and WRN-28-10 (Zagoruyko & Komodakis, 2016) as the backbone architectures. All methods were trained on the training set of the ID data, and after training, we combined the ID test set with the OOD data for evaluation.

### 4.1 Comparison with Unsupervised Methods

We initially compared the performance of our method with various unsupervised approaches by following the benchmark setting (Graham et al., 2023) as it requires only pseudo-labels. Specifically, we employed two distinct configurations: one for grayscale datasets and another for color datasets. In the grayscale setting, we used FashionMNIST (Xiao et al., 2017) as the ID dataset and MNIST as the OOD dataset. For the color setting, we evaluated three datasets: CIFAR-10 (Krizhevsky et al., 2009), CelebA (Liu et al., 2015), and SVHN (Netzer et al., 2011). Each dataset was treated as the ID dataset in turn, with the other two serving as OOD datasets. Additionally, we included vertically and horizontally flipped versions of each ID dataset as extra OOD datasets.

The AUROC values (Fawcett, 2006) in Table 3 demonstrate the effectiveness of our method. Given the variability in handling OOD samples, no baseline method consistently outperformed the others across all experiments. To address this, we computed the the average ranking of each method. A lower rank corresponds to better performance. Our method was ranked 1.8 across various experiment settings. The work of (Kamkari et al., 2024) was excluded from the rankings due to missing results in several settings. However, in the settings where their results were available, our method and (Kamkari et al., 2024) performed competitively in terms of detection accuracy. It is important to note that our method is based on a WideResNet(Zagoruyko & Komodakis, 2016) classifier, while (Kamkari et al., 2024) utilized a pre-trained normalizing flow Kingma & Dhariwal (2018); Durkan et al. (2019)

| Method | SVHN | | Textures | | LSUN-C | | LSUN-R | | iSUN | | Places365 | | Average | | Avg. Rank | |
|---|---|---|---|---|---|---|---|---|---|---|---|---|---|---|---|---|
| | FPR↓ | AUR↑ | FPR↓ | AUR↑ | FPR↓ | AUR↑ | FPR↓ | AUR↑ | FPR↓ | AUR↑ | FPR↓ | AUR↑ | FPR↓ | AUR↑ | FPR↓ | AUR↑ |
| MSP (Hendrycks & Gimpel, 2016) | 52.12 | 92.20 | 59.47 | 89.56 | 32.83 | 95.62 | 48.35 | 93.07 | 50.30 | 92.58 | 60.70 | 88.42 | 50.63 | 91.91 | 7.71 | 7.00 |
| ODIN (Liang et al., 2017) | 33.83 | 93.03 | 45.49 | 90.01 | 7.29 | 98.62 | 20.05 | 96.56 | 23.09 | 96.01 | 45.06 | 89.86 | 29.14 | 94.02 | 3.71 | 3.71 |
| EN (Liu et al., 2020) | 30.47 | 94.05 | 45.83 | 90.37 | 7.21 | 98.63 | 23.62 | 95.93 | 27.14 | 95.34 | 43.67 | 90.29 | 29.66 | 94.10 | 3.86 | 3.43 |
| EN+REACT (Sun et al., 2021) | 40.54 | 90.54 | 48.61 | 88.44 | 15.12 | 96.86 | 27.01 | 94.74 | 30.57 | 93.95 | 44.99 | 89.37 | 34.47 | 92.32 | 5.71 | 6.57 |
| EN+DICE (Sun & Li, 2022) | 25.95 | 94.66 | 47.22 | 89.82 | 3.83 | 99.26 | 27.70 | 95.01 | 31.07 | 94.42 | 49.28 | 88.08 | 30.84 | 93.54 | 5.00 | 4.57 |
| DML+ (Zhang & Xiang, 2023) | 33.16 | 90.59 | 27.01 | 92.86 | 5.98 | 98.71 | 48.72 | 88.89 | 23.41 | 94.89 | 47.01 | 87.78 | 30.88 | 92.45 | 4.71 | 5.57 |
| FN (Yu et al., 2023) | 7.13 | 98.65 | 31.18 | 92.31 | 0.07 | 99.96 | 27.08 | 95.25 | 26.02 | 95.38 | 62.54 | 84.62 | 25.67 | 94.36 | 3.29 | 3.00 |
| Our method | 22.82 | 93.91 | 25.18 | 94.25 | 1.84 | 99.61 | 32.47 | 93.98 | 6.71 | 98.55 | 32.38 | 92.63 | 20.23 | 95.4 | 2.00 | 2.14 |
| LINe(Ahn et al., 2023) (DNet-101) | 12.49 | 97.61 | 25.83 | 94.56 | 0.43 | 99.80 | - | - | 5.27 | 99.02 | 50.94 | 89.63 | 19.99 | 96.124 | - | - |
| SNN(Ghosal et al., 2024) (DNet-101) | 2.67 | 99.52 | 5.22 | 99.14 | 9.70 | 98.35 | 8.90 | 98.44 | 19.84 | 96.51 | 43.62 | 90.98 | 15.00 | 97.16 | - | - |
| Our method (WRN-28-10) | 4.36 | 97.22 | 3.85 | 99.16 | 1.76 | 99.60 | 18.67 | 96.64 | 12.06 | 97.63 | 22.27 | 95.07 | 12.16 | 97.55 | - | - |

Table 4: We compared our approach with supervised methods using the benchmark setting, utilizing ResNet18 as the backbone in the upper section and different backbones (D-NetHuang et al. (2017) and WRN-28-10) in the lower section of the table. In this experiment, the ID dataset was CIFAR-10, and the corresponding OOD datasets are shown in the first row. The results of the baseline methods were taken from (Yu et al., 2023; Ahn et al., 2023; Ghosal et al., 2024). We have highlighted the best results in bold and the second-best results are underlined.

or diffusion model Rombach et al. (2022) for density estimation, which would be time-consuming for OOD detection. In comparison to other baseline methods, our approach demonstrated superior performance on average, particularly in scenarios where OOD data consisted of vertically flipped ID images, with the exception of flipped FashionMNIST. While our method did not secure the top rank in every setting, it consistently remained competitive with the leading methods. An interesting observation from the AUROC values is that all methods faced difficulties in detecting OOD samples when they were horizontally flipped ID images. This result is understandable, as horizontally flipped images often represent the same object categories but captured from different perspectives.

## 4.2 COMPARISON WITH SUPERVISED METHODS

For supervised benchmarking, we followed the experiment outlined in (Yu et al., 2023), using the CIFAR-10 dataset (Krizhevsky et al., 2009) as the ID dataset, and evaluated the methods with six OOD datasets: SVHN (Netzer et al., 2011), Textures (Cimpoi et al., 2014), LSUN-C (crop) (Yu et al., 2015), LSUN-R (resize), iSUN (Xu et al., 2015), and Places365 (Zhou et al., 2017).

The upper part of Table 4 presents the OOD detection results using the ResNet18 backbone. In this experiment, we also computed the FPR95 (Liang et al., 2017) to assess the model's performance in high-recall scenarios. As shown, our method exceled in more complex settings with OOD datasets such as Textures, iSUN, and Places365, achieving the highest AUROC and lowest FPR95 values. On SVHN and LSUN-crop, it ranked second, slightly behind the method of (Yu et al., 2023). Notably, our method was ranked 2.14 across the experiment settings. It improved the average FPR95 by 21% and the AUROC by 1% compared to the second-best results. In the lower section of Table 4, our approach, trained on WRN-28-10, demonstrated competitive performance compared to state-of-the-art methods trained on a considerably larger backbone—DenseNet (Huang et al., 2017). It is important to note that we are comparing our unsupervised method against supervised methods.

**Comparison under Different Backbones.** We evaluated the performance of our method using the VGG11 and WideResNet architectures for CIFAR-10 and ReseNet18 for CIFAR-100. In this experiment, CIFAR-10 and CIFAR-100 were used as the ID datasets, with the same OOD datasets as those in Table 4. Table 5 presents the averaged scores across the OOD datasets, where our method once again outperforms the baseline methods, achieving the best overall performance. For detailed values of the results, we refer readers to our supplemental material.

## 4.3 DISCUSSIONS

**Ground-Truth Labels vs. Pseudo Labels.** The primary advantage of our method over approaches using GT labels, such as FeatureNorm Yu et al. (2023), likely stems from the use of SSL pre-trained weights rather than the pseudo-labels being inherently superior to GT labels. To validate

| Method | CIFAR-100 | | CIFAR-10 | | | |
| | ResNet18 | | WRN-28-10 | | VGG11 | |
| | FPR95↓ | AUROC↑ | FPR95↓ | AUROC↑ | FPR95↓ | AUROC↑ |
|---|---|---|---|---|---|---|
| MSP(Hendrycks & Gimpel, 2016) | 73.02 | 82.43 | 41.49 | 91.84 | 64.77 | 88.73 |
| ODIN(Liang et al., 2017) | 63.87 | 85.94 | 29.22 | 90.95 | 47.52 | 91.56 |
| EN(Liu et al., 2020) | 71.45 | 84.96 | 28.65 | 91.99 | 46.46 | 91.67 |
| EN+REACT(Sun et al., 2021) | 70.95 | 84.79 | 86.22 | 65.78 | 47.15 | 88.44 |
| EN+DICE(Sun & Li, 2022) | 70.78 | 85.17 | 31.53 | 89.30 | 50.80 | 90.98 |
| DML+(Zhang & Xiang, 2023) | 75.79 | 81.23 | 32.13 | 90.88 | 58.25 | 87.06 |
| FN(Yu et al., 2023) | 60.27 | 84.09 | 13.53 | 97.33 | 39.34 | 91.18 |
| Our method | **54.92** | **86.21** | **12.16** | **97.55** | **36.41** | **91.83** |

Table 5: We evaluated our method across different network architectures. The ID dataset was either CIFAR-10 or CIFAR-100, while the OOD datasets included SVHN, Textures, LSUN-C, LSUN-R, iSUN, and Places365. Due to space constraints, we present only the average AUROC and FPR95 values here. For detailed results, we refer readers to our supplemental material.

| Method | SVHN | | Textures | | LSUN-C | | LSUN-R | | iSUN | | Places365 | | Average | |
| | FPR↓ | AUR↑ | FPR↓ | AUR↑ | FPR↓ | AUR↑ | FPR↓ | AUR↑ | FPR↓ | AUR↑ | FPR↓ | AUR↑ | FPR↓ | AUR↑ |
|---|---|---|---|---|---|---|---|---|---|---|---|---|---|---|
| SSL+GT | **11.50** | **97.86** | **3.36** | **99.30** | **1.50** | **99.67** | 19.74 | 96.57 | **10.87** | **97.89** | 22.18 | **95.17** | **11.5** | **97.74** |
| SSL+PS | 14.36 | 97.22 | 3.85 | 99.16 | 1.76 | 99.60 | 18.67 | **96.64** | 12.06 | 97.63 | 22.27 | 95.07 | 12.16 | 97.55 |

Table 6: Performance comparison of fine-tuning SSL model using GT and pseudo (PS) labels.

this hypothesis, we conducted an additional experiment where we fine-tuned the model pre-trained with Mixed Barlow Twins using both GT labels and pseudo-labels. The results, presented in Table 6, show that the model fine-tuned with GT labels achieves better OOD detection performance. However, the performance difference between the two approaches is relatively small, which highlights the effectiveness of pseudo-labels. It is important to note that the use of GT labels for fine-tuning requires manually annotated data, which incurs additional costs.

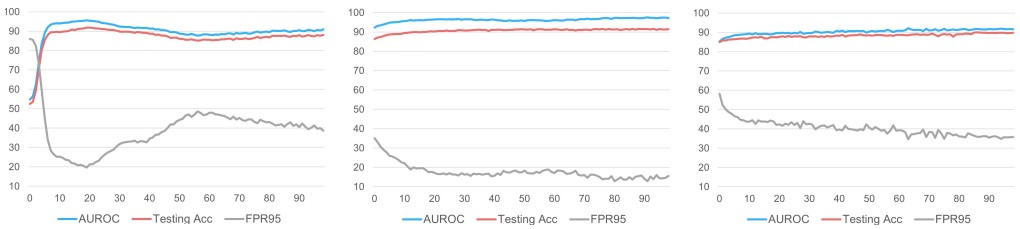

Figure 2: The experimental results using ResNet18, WRN-28-10, and VGG11 backbones show a strong correlation between OOD detection performance and classification accuracy on the ID test set.

**Model Generalization.** Model generalization is a crucial aspect of our method because we train a network to classify ID data into specific categories, which presents challenges when handling OOD data. If the network memorizes data categories rather than learning to classify, it will fail to distinguish between unseen ID and OOD data. To validate this assumption, we conducted an experiment using the supervised benchmark setting with ResNet18, WRN-28-10, and VGG11 as backbones. Figure 2 illustrates the relationship between OOD detection performance and classification accuracy on the test dataset. As shown, OOD detection performance was strongly correlated with the model's generalization ability, confirming our assumption across all three backbones. These results support our model selection strategy, where the network with the highest test accuracy was chosen to detect OOD samples.

**Number of Pseudo-Categories.** We applied the K-means clustering method to generate pseudo-labels for training the classifier. The classifier's performance in detecting OOD samples can vary depending on the number of pseudo-labels used for an ID dataset. To explore the impact of the

| Number | CIFAR-10 | | | | | CIFAR-100 | | | | |
|---|---|---|---|---|---|---|---|---|---|---|
| | Train | Test | Gap | FPR95 ↓ | AUROC ↑ | Train | Test | Gap | FPR95 ↓ | AUROC ↑ |
| K = 5 | 99.8 | 96.7 | 3.1 | 22.18 | 94.62 | - | - | - | - | - |
| K = 10 | 98.5 | 95.1 | 3.4 | 24.46 | 94.01 | 94.0 | 88.7 | 5.3 | 68.23 | 74.50 |
| K = 20 | 96.7 | 93.8 | **2.9** | **20.23** | **95.40** | 92.7 | 87.1 | 5.6 | 67.15 | 76.28 |
| K = 50 | 94.0 | 89.3 | 4.7 | 43.28 | 90.08 | 90.5 | 85.4 | 5.1 | 61.78 | 79.26 |
| K = 100 | 91.0 | 86.7 | 4.3 | 41.56 | 87.96 | 88.9 | 84.6 | **4.3** | 56.28 | 85.71 |
| K = 200 | 89.1 | 80.7 | 8.4 | 60.42 | 79.02 | 83.5 | 79.0 | 4.5 | **54.92** | **86.21** |
| K = 500 | - | - | - | - | - | 58.7 | 53.2 | 5.5 | 60.72 | 78.59 |

Table 7: We assessed the impact of the number of pseudo-categories using a supervised benchmark setting. The ID dataset was set to CIFAR-10 and CIFAR-100. As indicated, the gap in classification accuracy between the ID training and ID testing datasets can be used to determine the optimal number of pseudo-categories.

| Metric | GT Labels | | | | Pseudo Labels | | | |
|---|---|---|---|---|---|---|---|---|
| | **Best** | **Avg** | **Max** | **Min** | **Best** | **Avg** | **Max** | **Min** |
| **AUROC** | 93.79 | 94.01 (+0.22) | 93.18 (-0.61) | 94.29 (+0.50) | 95.4 | 95.83 (+0.43) | 94.68 (+0.28) | **96.35** (+1.95) |
| **FPR95** | 26.32 | 26.08 (-0.24) | 27.13 (+0.81) | 25.85 (-0.47) | 20.23 | 18.84 (-1.49) | 21.83 (+1.6) | **17.85** (-2.38) |

Table 8: We ensembled three models trained on GT labels and three models trained on pseudo-labels to compare their OOD detection performance. We reported the best performance from individual models, as well as the average, minimum, and maximum results from the ensemble. The values in parentheses indicate the improvement of the ensemble compared to the best-performing individual model. Notably, while the top-performing model trained on pseudo-labels surpassed the best model trained on GT labels, ensembling the pseudo-labeled models yielded an even greater improvement.

number of clusters, $k$, we conducted an experiment by following the setting used for backbone evaluation and analyzed the relationship between training and testing accuracy gaps.

The results, summarized in Table 7, reveal that the choice of $k$ is closely tied to the dataset's complexity. Interestingly, the model achieves near-optimal OOD detection performance when the number of pseudo-categories matches the GT categories. However, doubling the GT categories led to the best overall performance, suggesting that a higher number of pseudo-categories may better capture dataset complexity. On the other hand, significantly increasing the number of pseudo-categories diminishes the semantic distinction between them, hindering the model's ability to learn a generalizable classification logic and negatively impacting OOD detection performance. This is reflected in the relationship between the training-testing accuracy gap (measured as the L1 distance) and the model's generalization ability. A larger gap indicates weaker generalization, highlighting the model's struggle to capture meaningful classification logic. While the accuracy gap alone cannot pinpoint the optimal number of pseudo-categories, it serves as a valuable guideline for selecting a reasonable number of clusters to balance performance and generalization..

**Ensembling Pseudo-Labels.** We explored the effect of ensembling in our method, as the pseudo-labels generated from different clustering processes can vary, while the GT labels remain unique. In this case, the ensemble networks differ not only in their initial seeds but also in the categories they learn, potentially offering diverse perspectives beyond networks trained with GT labels. To assess this advantage, we repeated the supervised benchmark experiment by training three networks on the CIFAR-10 dataset with 5, 10, and 20 pseudo-categories. These three configurations were selected due to their similar detection performance, as shown in Table 7. To ensemble the results, we calculated the average, minimum, and maximum of the feature norms across channels to evaluate OOD detection performance. Recall from Section 3.4 that $z \in \mathbb{R}^{C \times W \times H}$ represents a feature map. We compute the feature norm for channel $c$ in model $i$ as:

$$N_i^c = \sqrt{\sum_{w=1}^{W} \sum_{h=1}^{H} \max(z_c(w,h), 0)^2}. \tag{5}$$

Let $N_i \in \mathbb{R}^{C \times 1 \times 1}$ be a norm vector consisting of $N_i^c$. The ensemble results are computed as:

$$N_{\text{avg}} = \frac{1}{k} \sum_{i=1}^{k} N_i, \qquad N_{\text{max}} = \max_{i=1}^{k} N_i, \qquad N_{\text{min}} = \min_{i=1}^{k} N_i, \qquad (6)$$

where $k = 3$ is the number of ensemble models, and the max and min functions operate element-wise. We then calculate the norm values $|N_{\text{avg}}|$, $|N_{\text{max}}|$, and $|N_{\text{min}}|$ to differentiate between ID and OOD samples. As in (Yu et al., 2023), higher norm values imply ID samples, while lower norm values indicate OOD samples.

As shown in Table 8, ensembling based on pseudo-labels outperformed ensembling with GT labels due to the increased model perspectives. Among all the ensemble strategies, the minimum ensemble delivered the best performance. This result is intuitive, as models tend to produce larger feature norm values for ID samples due to the familiarity of the data's structure Park et al. (2023). In contrast, OOD samples generate smaller feature norm values. By focusing on the minimum value ensemble, the smaller norms of OOD samples are emphasized, making them more distinguishable from the larger norms of ID data. This strategy reduces false positives and improves OOD detection performance compared to other ensemble approaches.

**Extended Experimental Evaluation** In addition to the main results demonstrating the efficacy of our method, we conducted further experiments to explore various factors. These include the OOD detection performance of the SSL method with and without the additional fine-tuning step (A.1), the choice of SSL embedding and clustering methods (A.2), the selection of baseline OOD detection methods (A.3), the extension of our approach to the larger ImageNet dataset (A.4), the evaluation of pseudo-labels generated using different SSL backbones (A.5), and the quality of the generated pseudo-labels (A.6) in the appendix. These additional experiments provide a more comprehensive understanding of our method and its versatility across diverse scenarios.

**Limitations.** Experiment results showed that our method slightly outperformed (Yu et al., 2023). Given that the main distinction between the two approaches is the use of pseudo-labels versus GT labels, we suspect that pseudo-labels generated by SSL may, in certain scenarios, offer advantages over GT labels. However, this outcome is not yet fully understood, and we cannot guarantee that our method will consistently outperform (Yu et al., 2023) across all experimental settings. Our future work will focus on investigating the reasons behind this result and further refining the generation of pseudo-labels.

## CONCLUSION

In this study, we bridge the gap between supervised and unsupervised OOD detection methods by training models to classify data into pseudo-categories. Our key insight is that the success of supervised methods relies on a classification strategy that may not align with real-world understanding. While supervised methods detect OOD samples by recognizing the difficulty in classifying unfamiliar data, this challenge mirrors what unsupervised methods encounter, as they also struggle to reconstruct OOD data due to the absence of prior exposure. To implement this idea, we employ self-supervised learning (SSL) feature extraction combined with clustering to generate pseudo-labels for training, creating a robust OOD detection framework. We demonstrate the advantages of this method over state-of-the-art supervised and unsupervised approaches in terms of detection accuracy, efficiency, and scalability in real-world applications.

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

# A APPENDIX

## A.1 FINE-TUNING THE SELF-SUPERVISED LEARNING MODEL USING PSEUDO LABELS

We train the classifier by fine-tuning a network that has been pre-trained using SSL. Since SSL can effectively extract features with semantic information, it is important to investigate whether fine-tuning the model with pseudo-labels is necessary. To answer this question, we compared the OOD detection performance of the model with and without this additional fine-tuning step. The results presented in Table 9 show that the model trained with pseudo-labels outperforms the one without them in OOD detection. This finding is not surprising, as we employ FeatureNorm Yu et al. (2023), a supervised method for detecting OOD samples, which inherently requires the model to accurately learn to ID data.

| Method | SVHN | | Textures | | LSUN-C | | LSUN-R | | iSUN | | Places365 | | Average | |
|---|---|---|---|---|---|---|---|---|---|---|---|---|---|---|
| | FPR↓ | AUR↑ | FPR↓ | AUR↑ | FPR↓ | AUR↑ | FPR↓ | AUR↑ | FPR↓ | AUR↑ | FPR↓ | AUR↑ | FPR↓ | AUR↑ |
| SSL | 24.31 | 92.83 | 42.80 | 87.74 | 20.16 | 94.45 | 78.57 | 73.33 | 89.03 | 64.11 | 74.29 | 72.95 | 54.86 | 80.9 |
| SSL+PS | **22.82** | **93.91** | **25.18** | **94.25** | **1.84** | **99.61** | **32.47** | **93.98** | **6.71** | **98.55** | **32.38** | **92.63** | **20.23** | **95.4** |

Table 9: Comparison of OOD detection performance with and without fine-tuning a pre-trained SSL network using pseudo-labels. In this table, "PS" stands for pseudo-labels.

## A.2 THE CHOICE OF SSL EMBEDDING AND CLUSTERING METHODS

The SSL embeddings of an ID dataset play an important role in generating pseudo-labels. While there are many available SSL methods presented, we compared the performance of SimCLR Chen et al. (2020), BYOL Grill et al. (2020), W-MSE Ermolov et al. (2021), and our chosen method, mix-BarlowTwin (Gedara Chaminda Bandara et al., 2023), with k-means clustering dividing the CIFAR-10 dataset into 20 clusters. The results in Table 10 indicate that mix-BarlowTwin yields the best OOD detection performance, likely due to its ability to learn more informative representations. This finding aligns with the KNN test accuracy, as the KNN accuracy values of SimCLR, BYOL, W-MSE were 87%, 88%, and 88%, respectively. Namely, the higher KNN accuracy for SSL methods corresponds to improved OOD detection performance.

The lower half of Table 10 investigates the use of different clustering algorithms, with mix-BarlowTwin as the SSL method to generate pseudo-labels from CIFAR-10. Specifically, we compared two widely used clustering methods, HDBSCAN and DBSCAN. The results suggest that pseudo-labels derived from these clustering techniques did not significantly improve OOD detection performance. We attribute this to the way HDBSCAN and DBSCAN treat boundary samples as unknowns, preventing these samples from being forced into specific categories. Despite testing various clustering parameters, approximately 30% of the data samples were classified as unknown, which reduced the number of samples available for training the classifier and ultimately lowered OOD detection performance.

| Method | SVHN | | Textures | | LSUN-C | | LSUN-R | | iSUN | | Places365 | | Avg. | | Acc |
|---|---|---|---|---|---|---|---|---|---|---|---|---|---|---|---|
| | FPR↓ | AUR↑ | FPR↓ | AUR↑ | FPR↓ | AUR↑ | FPR↓ | AUR↑ | FPR↓ | AUR↑ | FPR↓ | AUR↑ | FPR↓ | AUR↑ | |
| SIMCLR Chen et al. (2020) | 32.68 | 90.75 | 29.33 | 91.35 | 5.67 | 97.29 | 37.15 | 93.92 | 36.40 | 92.62 | 38.72 | 91.87 | 29.99 | 92.96 | 87% |
| BYOL Grill et al. (2020) | 26.21 | 93.06 | 26.38 | 92.28 | 6.87 | 97.58 | 24.18 | 95.81 | 35.75 | 92.78 | 43.54 | 90.50 | 27.15 | 93.67 | 88% |
| W-MSE Ermolov et al. (2021) | 28.37 | 92.24 | 26.66 | 92.62 | 4.91 | 97.11 | 27.40 | 93.49 | 36.75 | 92.71 | 40.60 | 90.93 | 27.45 | 93.18 | 88% |
| HDBSCAN McInnes et al. (2017) | **19.06** | **94.48** | 43.73 | 84.49 | 25.67 | 90.66 | 77.28 | 71.78 | 25.56 | 90.40 | 65.48 | 74.79 | 42.80 | 84.43 | - |
| DBSCAN Ester et al. (1996) | 21.15 | 93.47 | 53.17 | 79.17 | 31.84 | 86.99 | 76.94 | 70.99 | 28.12 | 88.71 | 66.85 | 73.75 | 46.34 | 82.17 | - |
| Ours (mBT + k-means) | 22.82 | 93.91 | **25.18** | **94.25** | **1.84** | **99.61** | **32.47** | **93.98** | **6.71** | **98.55** | **32.38** | **92.63** | **20.23** | **95.4** | **91%** |

Table 10: We evaluated the OOD detection performance of pseudo-labels generated using various SSL (top) and clustering methods (middle). Our results are shown at the bottom.

### A.3 THE BASELINE METHOD FOR USING PSEUDO LABELS

We chose to implement our idea using FeatureNorm Yu et al. (2023), which is currently the leading state-of-the-art method for supervised OOD detection. Our decision is based on three key factors: 1) its superior performance, 2) its reproducibility, and 3) its strong theoretical foundations Park et al. (2023). To support this choice, we conducted additional experiments using other supervised OOD detection methods in conjunction with our pseudo-labels. The results, shown in Table 11, reveal that these alternative methods performed worse than FeatureNorm in terms of OOD detection accuracy. We believe this further confirms that FeatureNorm is a fitting benchmark for our framework.

| Method | SVHN | | Textures | | LSUN-C | | LSUN-R | | iSUN | | Places365 | | Average | |
|---|---|---|---|---|---|---|---|---|---|---|---|---|---|---|
| | FPR↓ | AUR↑ | FPR↓ | AUR↑ | FPR↓ | AUR↑ | FPR↓ | AUR↑ | FPR↓ | AUR↑ | FPR↓ | AUR↑ | FPR↓ | AUR↑ |
| EnergyLiu et al. (2020) | 33.43 | 92.98 | 42.66 | 90.51 | 18.98 | 97.35 | 28.33 | 93.29 | 28.18 | 93.31 | 54.22 | 88.19 | 34.3 | 92.60 |
| MahalanobisLee et al. (2018) | 28.39 | 93.98 | 23.56 | 94.2 | 4.04 | 99.19 | 38.89 | 92.64 | 16.08 | 96.77 | 48.45 | 91.44 | 26.57 | 94.70 |
| DML+ (Zhang & Xiang, 2023) | 45 | 82.06 | 17.95 | 90.58 | 24.19 | 94.51 | 39.35 | 87.68 | 35.35 | 86 | 33.11 | 86.13 | 32.49 | 87.82 |
| MSP (Hendrycks & Gimpel, 2016) | 17.94 | 88.53 | 32.91 | 89.87 | 18.65 | 89.36 | 88.24 | 61.83 | 46.04 | 88.03 | 54.48 | 85.62 | 43.3 | 83.87 |
| NAN (Park et al., 2023) | 35.62 | 91.47 | 75.5 | 86.25 | 2.81 | 99.37 | 40.15 | 99.28 | 16.60 | 95.75 | 35.66 | 92.17 | 34.6 | 93.10 |
| Ours (FeatureNorm) | 22.82 | 93.91 | 25.18 | 94.25 | 1.84 | 99.61 | 32.47 | 93.98 | 6.71 | 98.55 | 32.38 | 92.63 | 20.23 | 95.4 |

Table 11: We extended several baseline supervised OOD detection methods by incorporating pseudo-labels determined through self-supervised learning. We conducted experiments in accordance with the benchmark settings established in our paper. Our approach of combining FeatureNorm (Yu et al., 2023) with pseudo-labels yielded the best performance.

### A.4 MORE EVALUATIONS ON IMAGENET AS AN ID DATASET

We extended the evaluation on ImageNet as an ID Dataset using the ResNet-50 model. In this experiment, we utilized DINO Caron et al. (2021) as the SSL method, since it achieved the highest KNN accuracy on ImageNet.

The methods listed in the upper part of the Table 12 are supervised, whereas those in the lower part are unsupervised. Despite the inherent disadvantage of lacking GT labels, our method demonstrates competitive performance, achieving the highest scores on two datasets (SUN and Places) and performing strongly on others, securing the second-best average AUROC and FPR95. Notably, our unsupervised approach narrows the performance gap with supervised methods, reducing the average FPR by 24% and AUROC by 3%. These results underscore the effectiveness of our approach in OOD detection, even in the absence of GT labels, highlighting its practical utility.

| Method | INaturalist | | SUN | | Places | | Texture | | Average | |
|---|---|---|---|---|---|---|---|---|---|---|
| | FPR↓ | AUR↑ | FPR↓ | AUR↑ | FPR↓ | AUR↑ | FPR↓ | AUR↑ | FPR↓ | AUR↑ |
| MSP Hendrycks & Gimpel (2016) | 29.74 | 93.78 | 59.54 | 84.56 | 60.94 | 84.28 | 50.02 | 84.9 | 50.06 | 86.88 |
| EnergyLiu et al. (2020) | 20.98 | 96.17 | 47.05 | 88.91 | 51.15 | 87.7 | 39.31 | 88.9 | 39.62 | 90.42 |
| Maxlogit Hendrycks et al. (2019) | 22.06 | 95.99 | 50.9 | 88.43 | 53.78 | 87.37 | 42.25 | 88.42 | 42.25 | 90.05 |
| FN Yu et al. (2023) | **22.01** | **95.76** | 42.93 | 90.21 | 56.8 | 84.99 | **20.07** | **95.39** | **35.45** | **91.59** |
| SSD Sehwag et al. (2021) | 93.87 | 60.34 | 78.41 | 80.89 | 81.26 | 77.23 | 33.53 | 90.19 | 71.77 | 77.16 |
| KNN Sun et al. (2022) | 78.71 | 84.53 | 76.06 | 82.26 | 80.65 | 77.5 | 24.61 | 91.99 | 65.01 | 84.07 |
| NAN Park et al. (2023) | 36.09 | 92.9 | 56.27 | 86.76 | 65.08 | 83.22 | 46.86 | 87.57 | 51.08 | 87.61 |
| Ours | 28.56 | 93.46 | **42.81** | **90.93** | **56.67** | **83.58** | 27.25 | 93.21 | 38.82 | 90.35 |

Table 12: We assess the OOD detection performance using ImageNet as the ID dataset. The first row lists the OOD datasets. The methods in the upper section are supervised, utilizing ResNet-50 as their backbone, while those in the lower section are unsupervised. The baseline method results are taken from the papers by Yu et al. (2023) and Park et al. (2023). The best model is highlighted in bold.

### A.5 EVALUATION OF SSL BACKBONES

We compare the performance of pseudo-labels generated using two different backbones, ViT-S/8 and ResNet-50, for the ImageNet dataset. Both backbones were trained using the DINO SSL method, and we used ResNet-50 as the classifier in subsequent OOD detection experiments. Table 13 presents the results of these experiments, showing that pseudo-labels generated with ViT-S/8, which has higher KNN accuracy, outperform those generated with ResNet-50. This aligns with our earlier

explanations (section 4.3 and A.4) and further highlights the importance of the learned feature space and its embeddings in the success of our method.

| Method | INaturalist | | SUN | | Places | | Texture | | Average | | KNN accuracy |
|--------|------|------|------|------|------|------|------|------|------|------|--------------|
| | FPR↓ | AUR↑ | FPR↓ | AUR↑ | FPR↓ | AUR↑ | FPR↓ | AUR↑ | FPR↓ | AUR↑ | |
| ResNet-50 | 29.25 | 93.25 | 43.72 | 89.62 | 58.4 | 83.15 | 27.1 | 93.41 | 39.62 | 89.85 | 75.3% |
| VIT(S/8) | **28.56** | **93.46** | **42.81** | **90.93** | **56.67** | **83.58** | **27.25** | **93.21** | **38.82** | **90.35** | **79.7%** |

Table 13: We assess the OOD detection performance using ImageNet as the ID dataset. The first row lists the OOD datasets. The pseudo labels are generated through different SSL backbones and trained on Resnet-50 classifier

## A.6 Quality of Pseudo labels.

We train a classifier using pseudo-labels to achieve OOD detection. As shown in the pilot study, pseudo-labels generated through SSL outperform those created via raw data clustering or random assignment. In this section, we further demonstrate how different stages of SSL training can influence model performance. To illustrate this, we compared pseudo-labels generated at various stages of SSL training, obtained the KNN (Cover & Hart, 1967) accuracy of the model, and followed the supervised benchmark settings to measure the effect on OOD detection's performance. The results in Table 14 demonstrate a clear trend: as the quality of pseudo-labels improves, OOD detection performance also increases. This indicates that a network can learn meaningful classification rules as long as the dataset is partitioned effectively. Our findings indicate a connection between evaluation accuracy and OOD detection performance, offering useful insights into model behavior.

| Method | FPR ↓ | AUR ↑ |
|--------|-------|-------|
| SSL@20% KNN accuracy | 80.43 | 66.83 |
| SSL@50% KNN accuracy | 55.68 | 79.16 |
| SSL@91% KNN accuracy | 20.23 | 95.4 |

Table 14: The quality of pseudo-labels has a significant impact on OOD detection accuracy. As the SSL accuracy improves, the detection performance also increases accordingly.

