# OpenReview forum: "Pseudo-Labels are All You Need for Out-Of-Distribution Detection"
_ICLR.cc/2025/Conference — Submitted to ICLR 2025_

### Official Review · Reviewer_rnUz · 2024-10-21

**Soundness:** 2
**Presentation:** 3
**Contribution:** 2
**Rating:** 3
**Confidence:** 4

**Summary:**

This paper proposed an unsupervised OOD detection method using pseudo-categories generation and self-supervised learning, which achieved quite good performance in the OOD benchmark for CIFAR10. The theory is interesting and point out a critical problem: "Does a specific classification strategy necessarily align with real-world understanding? "

**Strengths:**

The authors propose the use of pseudo-categories generation and self-supervised learning to enhance the unsupervised OOD detection.

The idea is well reasoned with supporting experiments.

The authors propose framework with different backbone to improve performance through combination of  self-supervised training and supervised pseudo-classification learning.

The paper is well written and easy to follow.

**Weaknesses:**

I reckon there are several critical weaknesses in this paper.

1. The paper does not correctly point out the weaknesses of supervised and unsupervised OOD detection. In line 43-44, the reliance on extensive human-labeled data is not the main weakness for all supervised methods, but only for methods based on outliers exposure. Similarly, in line 47-48, "slow" and "complexity of models" are not the main limitations for unsupervised OOD detection. The objective of this paper is not clear.

2. A deeper investigation of why and how self-supervised learning happen must be provided. The main manuscript does not provide details and evaluations for self-supervised learning, which seems to be important in the proposed framework (correct me if I am wrong). Moreover, the explanation preceding the method is very difficult to understand, although the method itself is straightforward.

3. The ablation study is not sufficient. The ablations on self-supervised learning task and other clustering methods should be provided.

4. The results are not valid enough. The main results are only based on the OOD detection performance on CIFAR-10 (in-distribution dataset). At least, the experiment should be applied on more in-distribution datasets like CIFAR -100, minst, and imagenet. See :
 J. Yang, P. Wang, D. Zou, Z. Zhou, K. Ding, W. Peng, H. Wang, G. Chen, B. Li, Y. Sun, et al. Openood: Benchmarking generalized out-of-distribution detection. Advances in Neural Information Processing Systems, 35:32598–32611, 2022.

**Questions:**

See weaknesses.

---

> ### Author Response · Authors · 2024-11-20
> **We appreciate the reviewer's insightful comments. Below, we address the questions(1 & 2) raised by the reviewer.(Part 1)**
>
> ## Reviewer rnUz
>
> ### Weaknesses
>
> **Q1:**
> The paper does not correctly point out the weaknesses of supervised and unsupervised OOD detection. In line 43-44, the reliance on extensive human-labeled data is not the main weakness for all supervised methods, but only for methods based on outliers exposure. Similarly, in line 47-48, "slow" and "complexity of models" are not the main limitations for unsupervised OOD detection. The objective of this paper is not clear.
>
> **A1:**
> We acknowledge that different supervised methods may have varying degrees of reliance on human-labeled data. However, compared to our approach, supervised methods indeed require labeled data, which can become a limiting factor in many cases. Additionally, most unsupervised methods, especially those based on sample reconstruction, involve higher computational overhead. This is primarily due to the architectural differences: while supervised methods typically rely on an encoder, unsupervised methods often require both an encoder and a decoder for reconstruction. This computational demand is further exacerbated if the reconstruction process uses autoregressive techniques or iterative diffusion models. As shown in Table 1 of the manuscript, the supervised state-of-the-art method, FeatureNorm, is considerably faster than the unsupervised state-of-the-art method, DDPM, in identifying OOD samples.
>
> We would appreciate it if the reviewer could provide specific examples of what they consider the primary weaknesses of supervised and unsupervised methods, as this would greatly enhance our understanding of your viewpoint.
>
> ---
>
> **Q2:**
> A deeper investigation of why and how self-supervised learning happen must be provided. The main manuscript does not provide details and evaluations for self-supervised learning, which seems to be important in the proposed framework (correct me if I am wrong). Moreover, the explanation preceding the method is very difficult to understand, although the method itself is straightforward.
>
> **A2:**
> We appreciate the reviewer’s attention to the role of self-supervised learning (SSL) in our framework. Our implementation of SSL follows the mix-Barlow Twin method, which is based on a well-established approach. Given that our primary contribution lies in high-level conceptual innovation rather than in developing new SSL training methodologies, we did not include an extensive evaluation of the SSL component itself.
>
> The novelty of our work resides in our conceptual insight that supervised methods can detect OOD samples because they have learned a classification logic specific to the ID dataset, rather than genuinely recognizing real-world objects (such as cars or people). This classification logic enables the model to generalize this learned logic to testing data. Consequently, the difference in performance between ID and OOD datasets can effectively distinguish the two.
>
> Building on this insight, we employ SSL to generate pseudo-labels, thereby transforming supervised OOD detection methods into unsupervised ones. Although simple, this approach is far from intuitive; if it were, the longstanding bifurcation in OOD research into supervised and unsupervised categories likely would not exist. Our contribution is not in algorithmic novelty or model training, but rather in advancing a high-level concept that leverages pseudo-labels to train supervised methods for OOD detection. This approach combines the advantages of superior detection accuracy and faster detection speeds with the benefits of being annotation-free.

---

> > ### Author Response · Authors · 2024-11-20
> > **We appreciate the reviewer's insightful comments. Below, we address the question(3) raised by the reviewer.(Part 2)**
> >
> > **A3:**
> > Thank you for your suggestion to expand the ablation study. As we did not propose a new SSL method, our focus in this rebuttal is to compare the performance of pseudo-labels generated by different SSL techniques to assess their effectiveness in OOD detection. In the upper half of Table 1, we present a comparison among SimCLR, BYOL, W-MSE, and our chosen method, mix-BarrowTwin, with k-means clustering dividing the CIFAR-10 dataset into 20 clusters. The results indicate that mix-Barlow Twin yields the best OOD detection performance, likely due to its ability to learn more informative representations. This conclusion aligns with the KNN test accuracy, as well as with the findings in Table 7 of our manuscript, where higher KNN test accuracy for SSL methods corresponds to improved OOD detection performance.
> >
> > In the lower half of Table 1, we explore different clustering algorithms using mix-BarrowTwin as the SSL method to generate pseudo-labels from CIFAR-10. We compared HDBSCAN and DBSCAN, two widely used clustering methods. However, the results suggest that pseudo-labels derived from these clustering methods do not significantly improve OOD detection. We attribute this to HDBSCAN and DBSCAN treating boundary samples as unknowns, thus not forcing them into specific categories. Despite testing various clustering parameters, approximately 30% of data samples were classified as unknown, reducing the number of samples available for training the classifier and ultimately lowering OOD detection performance.
> >
> >
> > Table 1: We evaluated the OOD detection performance of pseudo-labels generated using various SSL(top) and clustering methods (middle). Our results are shown at the bottom
> > | Method                | SVHN FPR ↓ | SVHN AUR ↑ | Textures FPR ↓ | Textures AUR ↑ | LSUN-C FPR ↓ | LSUN-C AUR ↑ | LSUN-R FPR ↓ | LSUN-R AUR ↑ | iSUN FPR ↓ | iSUN AUR ↑ | Places365 FPR ↓ | Places365 AUR ↑ | Avg. FPR ↓ | Avg. AUR ↑ | Acc |
> > |-----------------------|------------|------------|----------------|----------------|--------------|--------------|--------------|--------------|------------|------------|-----------------|-----------------|------------|------------|-----|
> > | SIMCLR [2]               | 32.68      | 90.75      | 29.33          | 91.35          | 5.67         | 97.29        | 37.15        | 93.92        | 36.40      | 92.62      | 38.72           | 91.87           | 29.99      | 92.96      | 87% |
> > | BYOL [5]                 | 26.21      | 93.06      | 26.38          | 92.28          | 6.87         | 97.58        | 24.18        | 95.81        | 35.75      | 92.78      | 43.54           | 90.50           | 27.15      | 93.67      | 88% |
> > | W-MSE [3]                | 28.37      | 92.24      | 26.66          | 92.62          | 4.91         | 97.11        | 27.40        | 93.49        | 36.75      | 92.71      | 40.60           | 90.93           | 27.45      | 93.18      | 88% |
> > |-|-|-|-|-|-|-|-|-|-|-|-|-|-|-|-|
> > | HDBSCAN [9]              | **19.06**     | **94.48**      | 43.73          | 84.49          | 25.67        | 90.66        | 77.28        | 71.78        | 25.56      | 90.40      | 65.48           | 74.79           | 42.80      | 84.43      | -   |
> > | DBSCAN  [4]              | 21.15      | 93.47      | 53.17          | 79.17          | 31.84        | 86.99        | 76.94        | 70.99        | 28.12      | 88.71      | 66.85           | 73.75           | 46.34      | 82.17      | -   |
> > | Ours (mBT + k-means)  | 22.82      | 93.91      | **25.18**          | **94.25**          | **1.84**         | **99.61**        | **32.47**        | **93.98**        | **6.71**       | **98.55**      | **32.38**           | **92.63**           | **20.23**      | **95.4**       | **91%** |

---

> ### Author Response · Authors · 2024-11-20
> **We appreciate the reviewer's insightful comments. Below, we address the question(4) raised by the reviewer.(Part 3)**
>
> **Q4:**
> The results are not valid enough. The main results are only based on the OOD detection performance on CIFAR-10 (in-distribution dataset). At least, the experiment should be applied on more in-distribution datasets like CIFAR-100, MNIST, and ImageNet. See: *J. Yang, et al., "OpenOOD: Benchmarking generalized out-of-distribution detection." NeurIPS 2022*.
>
> **A4:**
> Thank you for your feedback regarding the scope of our experiments. We would like to clarify that our study included multiple in-distribution (ID) datasets beyond CIFAR-10, as indicated in Table 3, where we also used FMNIST, CelebA, and SVHN as ID datasets (denoted in the first row). Furthermore, in Table 5, the four rightmost columns provide results using CIFAR-100 as the ID dataset.
>
> To address your suggestion, we have now extended our experiments to include ImageNet as an additional ID dataset. The results of this experiment are presented in Table 2 of this rebuttal. We hope these additions further strengthen the validity of our findings and provide a more comprehensive evaluation of our approach. In this experiment, we employed DINO [1] as the SSL method, as it achieved the highest KNN accuracy on ImageNet.
>
> The methods listed in the upper part of Table 2 are supervised, utilizing ground truth (GT) labels to train the model, whereas those in the lower part are unsupervised, relying solely on sample-level information during training. Despite the inherent disadvantage of lacking GT labels, our method demonstrates competitive performance, achieving the highest scores on two datasets (SUN and Places) and performing strongly on others, securing the second-best average AUROC and FPR95. Notably, our unsupervised approach narrows the performance gap with supervised methods, reducing the average FPR by 24\% and AUROC by 3\%. These results underscore the effectiveness of our approach in OOD detection, even in the absence of GT labels, highlighting its practical utility.
>
>
> Table 2: We assess the OOD detection performance using ImageNet as the ID dataset. The first row lists the OOD datasets. The methods in the upper section are supervised, utilizing ResNet-50 as their backbone, while those in the lower section are unsupervised. The baseline method results are taken from the papers by [10] and [13]. The best model is highlighted in bold.
> | Method     | INaturalist FPR ↓ | INaturalist AUR ↑ | SUN FPR ↓ | SUN AUR ↑ | Places FPR ↓ | Places AUR ↑ | Texture FPR ↓ | Texture AUR ↑ | Avg. FPR ↓ | Avg. AUR ↑ |
> |------------|-------------------|-------------------|-----------|-----------|--------------|--------------|----------------|---------------|------------|------------|
> | MSP  [6]      | 29.74            | 93.78            | 59.54     | 84.56     | 60.94        | 84.28        | 50.02         | 84.9          | 50.06      | 86.88      |
> | Energy  [8]   | 20.98            | 96.17            | 47.05     | 88.91     | 51.15        | 87.7         | 39.31         | 88.9          | 39.62      | 90.42      |
> | Maxlogit [7]  | 22.06            | 95.99            | 50.9      | 88.43     | 53.78        | 87.37        | 42.25         | 88.42         | 42.25      | 90.05      |
> | FN  [13]       | **22.01**        | **95.76**        | 42.93     | 90.21     | 56.8         | 84.99        | **20.07**     | **95.39**     | **35.45**  | **91.59**  |
> |-|-|-|-|-|-|-|-|-|-|-|
> | SSD  [11]      | 93.87            | 60.34            | 78.41     | 80.89     | 81.26        | 77.23        | 33.53         | 90.19         | 71.77      | 77.16      |
> | KNN   [12]     | 78.71            | 84.53            | 76.06     | 82.26     | 80.65        | 77.5         | 24.61         | 91.99         | 65.01      | 84.07      |
> | NAN   [10]     | 36.09            | 92.9             | 56.27     | 86.76     | 65.08        | 83.22        | 46.86         | 87.57         | 51.08      | 87.61      |
> | Ours       | 28.56            | 93.46            | **42.81** | **90.93** | **56.67**    | **83.58**    | 27.25         | 93.21         | 38.82      | 90.35      |

---

> ### Author Response · Authors · 2024-11-20
> **We appreciate the reviewer's insightful comments. Below are the references used in the response above (Part 4**
>
> **References:**
> 1. Mathilde Caron, Hugo Touvron, Ishan Misra, Hervé Jégou, Julien Mairal, Piotr Bojanowski, and Armand Joulin. *Emerging properties in self-supervised vision transformers*. In *Proceedings of the IEEE/CVF International Conference on Computer Vision*, pp. 9650–9660, 2021.
>
> 2. Ting Chen, Simon Kornblith, Mohammad Norouzi, and Geoffrey Hinton. *A simple framework for contrastive learning of visual representations*. In *International Conference on Machine Learning*, pp. 1597–1607. PMLR, 2020.
>
> 3. Aleksandr Ermolov, Aliaksandr Siarohin, Enver Sangineto, and Nicu Sebe. *Whitening for self-supervised representation learning*. In *International Conference on Machine Learning*, pp. 3015–3024. PMLR, 2021.
>
> 4. Martin Ester, Hans-Peter Kriegel, Jörg Sander, Xiaowei Xu, et al. *A density-based algorithm for discovering clusters in large spatial databases with noise*. In *KDD*, volume 96, pp. 226–231, 1996.
>
> 5. Jean-Bastien Grill, Florian Strub, Florent Altché, Corentin Tallec, Pierre Richemond, Elena Buchatskaya, Carl Doersch, Bernardo Avila Pires, Zhaohan Guo, Mohammad Gheshlaghi Azar, et al. *Bootstrap your own latent: A new approach to self-supervised learning*. In *Advances in Neural Information Processing Systems*, 33:21271–21284, 2020.
>
> 6. Dan Hendrycks and Kevin Gimpel. *A baseline for detecting misclassified and out-of-distribution examples in neural networks*. *arXiv preprint arXiv:1610.02136*, 2016.
>
> 7. Dan Hendrycks, Steven Basart, Mantas Mazeika, Andy Zou, Joe Kwon, Mohammadreza Mostajabi, Jacob Steinhardt, and Dawn Song. *Scaling out-of-distribution detection for real-world settings*. *arXiv preprint arXiv:1911.11132*, 2019.
>
> 8. Weitang Liu, Xiaoyun Wang, John Owens, and Yixuan Li. *Energy-based out-of-distribution detection*. In *Advances in Neural Information Processing Systems*, 33:21464–21475, 2020.
>
> 9. Leland McInnes, John Healy, Steve Astels, et al. *HDBSCAN: Hierarchical density-based clustering*. *Journal of Open Source Software*, 2(11):205, 2017.
>
> 10. Jaewoo Park, Jacky Chen Long Chai, Jaeho Yoon, and Andrew Beng Jin Teoh. *Understanding the feature norm for out-of-distribution detection*. In *Proceedings of the IEEE/CVF International Conference on Computer Vision*, pp. 1557–1567, 2023.
>
> 11. Vikash Sehwag, Mung Chiang, and Prateek Mittal. *SSD: A unified framework for self-supervised outlier detection*. *arXiv preprint arXiv:2103.12051*, 2021.
> 12. Yiyou Sun, Yifei Ming, Xiaojin Zhu, and Yixuan Li.*Out-of-distribution detection with deep nearest neighbors. In International Conference on Machine Learning, pp. 20827–20840. PMLR*, 2022.
> 13. Yeonguk Yu, Sungho Shin, Seongju Lee, Changhyun Jun, and Kyoobin Lee. *Block selection method for using feature norm in out-of-distribution detection. In Proceedings of the IEEE/CVF Conference on Computer Vision and Pattern Recognition, pp. 15701–15711*, 2023.

---

> ### Author Response · Authors · 2024-11-29
>
> Dear Reviewer rnUz,
>
> We appreciate the time and effort you have invested in reviewing our manuscript. If possible, could you kindly let us know if our responses have adequately addressed your questions? Please let us know if you have any further comments. We would be delighted to provide additional responses. Thank you.

---

### Official Review · Reviewer_cCFA · 2024-10-24

**Soundness:** 2
**Presentation:** 2
**Contribution:** 2
**Rating:** 5
**Confidence:** 3

**Summary:**

The paper presents a method to bridge the gap between supervised and unsupervised OOD detection methods by training models to classify data into pseudo categories. They employ a self supervised learning (SSL) technique for feature extraction, and used clustering to generate pseudo labels for training to realize robust OOD detection framework. Overall, the paper is written well and is well motivated by the fact that pseudo labels from high-level representations can improve the classification performance. The authors have also conducted significant amount of experiments and provided discussions on various test scenarios to validate their performance.

**Strengths:**

The paper is written well and easy to follow up. The motivation for detection OOD samples using a SSL technique is quite interesting. There are different types of experiments done to validate the proposed method such as comparison with different backbone networks, and using different types of OOD and ID datasets.

**Weaknesses:**

The novelty of the paper is not that convincing because they have integrated existing methods such as BT loss and mixup regularization into the training objective. In the introduction section, the main contribution is not clear. Although the paper provides motivation on how pseudo labels from high-level representation are necessary for effective OOD classification but this hypothesis is not well proved in the pilot study. The paper seems to use different methods such as a specific siamese network Barlow Twins architecture, but why is this used and why is this preferred over other architecture is not discussed. Also, the integration of BT loss with mixup regularization is not clear. The explanation saying "for the regularization" is not convincing. There is not enough evidence to support this, or the authors have not done any ablation on this to the least. Finally, the paper lacks the information about reproducibility. Not enough information regarding the algorithms and source codes are provide.

**Questions:**

1. What is cluster-based labels? It should be first described as the reader will not have knowledge about it in prior while reading the paper.
2. In Table. 2 for random label assignment, why the auroc is significantly higher than its test accuracy counterpart.? While for raw image clustering and GT labels, the difference between auroc and test accuarcy is not that significant?
2. In line 183, findings and analysis, what does certain methodology mean? There should be clear explanation of this method.
3. How is the method reliant on low-level pixel colors? There is no enough evidence to support this statement. Also not enough prior information is provided on this matter.
4. Based on the pilot study, it is not concluded that pseudo labels need to be derived from high-level representations. Instead it is concluded that, because existing classification networks uses the representations of input from a layer it performs better. Its not that, if pseudo labels are derived from these representations, the classification accuracy will increase. This explanation is naive and enough proof is not provided to support this hypothesis. There is not enough proof provided using the pseudo labels and the high-label representations.
5. In section, 3.3, it is said that Y^A and Y^B are passed through encoder and a projector. What is a projector? Is it a decoder?
6. What us Y^M here and where does it come from?
7. The schematics of the figure 1 is quite irrelevant to the explanation provided in section 3.3. For instance, what does dotted vertical line represent between two encoders. And Where is Y^M coming from ?

---

> ### Author Response · Authors · 2024-11-24
> **We appreciate the reviewer’s(cCFA) insightful comments. Below, we address the questions raised by the reviewer(Part 1)**
>
> ## Reviewer cCFA
>
> ### Weaknesses
>
> #### Q1:
>  The novelty of the paper is not that convincing because they have integrated existing methods such as BT loss and mixup regularization into the training objective. In the introduction section, the main contribution is not clear.
>
> #### A1:
> We appreciate your feedback. However, we contend that our insight - that supervised and unsupervised OOD detection methods are fundamentally similar - is indeed innovative. Specifically, they utilize ID data to learn a particular task. Supervised methods learn to classify ID data, while unsupervised methods learn to reconstruct it. These methods then rely on the performance discrepancy between ID and OOD data to identify OOD samples. Although simple, this approach is far from intuitive; if it were, it would be challenging to justify the longstanding bifurcation of research into supervised and unsupervised OOD detection pathways. The potential publication of our manuscript could inspire a merging of these research paths, marking a significant shift in the field of OOD detection methods in the future.
>
> We believe that novelty should not be confined to the development of base algorithms or training techniques alone. Our insight contributes at a conceptual level, where utilizing pseudo-labels allows us to train supervised methods without manual annotation, achieving higher detection accuracy and faster detection speeds. While pursuing algorithmic novelties along the two traditional paths might appear novel, it often lacks the practical benefits that our approach offers by combining the strengths of both supervised and unsupervised methodologies.
>
> ---
>
> #### Q2:
> Although the paper provides motivation on how pseudo labels from high-level representation are necessary for effective OOD classification but this hypothesis is not well proved in the pilot study.
>
> #### A2:
> We have revised the paragraph titled **Findings and Analysis**. The necessity of generating pseudo-labels through self-supervised learning (SSL) is particularly explained in the latter part of this paragraph, which has been highlighted in italics.
>
> Table 2 of manuscipt summarizes the results of this pilot study. The model trained with GT labels achieved the best performance, exhibiting the lowest false positive rate at 95\% true positive rate (FPR95) [2] and the highest area under the receiver operating characteristic curve (AUROC). The model trained with cluster-based labels from raw image clustering performed moderately, while the model trained with random labels showed the poorest performance. Notably, the models' OOD detection capabilities were consistent with their classification accuracy on the testing data: higher testing accuracy corresponded to better OOD detection performance.
>
> These findings highlight the critical role of label quality in training effective classification models for OOD detection. The model trained with random labels failed to learn meaningful classification rules and could not effectively distinguish between ID and OOD samples. The model trained with cluster-based labels from raw images performed better than random labeling but still fell short compared to the model trained with GT labels. *This is likely because clustering in the raw image space captures low-level features like pixel intensities and colors, which do not align well with the high-level semantic features that modern classification networks extract.*
>
> *The moderate performance of the raw image clustering model underscores the need for pseudo-labels derived from high-level representations. Since classification networks like ResNet18 reduce spatial resolution and focus on abstract features, labels based on high-level semantic information are more conducive to learning effective classification strategies. SSL methods are particularly well-suited for this purpose, as they can learn rich, high-level features without requiring labeled data.*
>
> *By deriving pseudo-labels from high-level representations obtained through SSL, we can capture the semantic content of the images more effectively. This approach is expected to improve both classification accuracy and OOD detection performance, bringing models trained with pseudo-labels closer to those trained with GT labels. Thus, the pilot study demonstrates the importance of using SSL pseudo-labels to enhance OOD detection capabilities.*

---

> > ### Author Response · Authors · 2024-11-24
> > **We appreciate the reviewer’s(cCFA) insightful comments. Below, we address the questions raised by the reviewer(Part 2)**
> >
> > #### Q3:
> > The paper seems to use different methods such as a specific siamese network Barlow Twins architecture, but why is this used and why is this preferred over other architecture is not discussed. Also, the integration of BT loss with mixup regularization is not clear. The explanation saying "for the regularization" is not convincing. There is not enough evidence to support this, or the authors have not done any ablation on this to the least.
> >
> > #### A3:
> > Thank you for your valuable suggestions and insights. We appreciate your request for a more detailed explanation and ablation study regarding the choice of SSL method.
> >
> > We selected the mix-Barlow Twin for its ability to extract high-quality representations and generate pseudo-labels effectively. This choice was guided by our pilot study results (Table 1), which demonstrated that the closer pseudo-labels are to capturing the true semantic structure of the data, the better the model performs in both classification and OOD detection tasks. By applying KNN classification to the learned representations using a small subset of data with ground-truth labels, we directly evaluated how well the SSL method captured the dataset's semantic structure. Higher KNN accuracy indicates more discriminative representations that closely align with true class distinctions.
> >
> > To address this concern, we have conducted additional experiments (Table 2), comparing several SSL approaches, including SimCLR, BYOL, W-MSE, and our selected mix-Barlow Twin. The results highlight that mix-Barlow Twin consistently outperforms the other methods due to its ability to achieve higher KNN accuracy, which reflects its superior capability to learn informative representations. These findings are consistent with those presented in Table 7 of our manuscript, where SSL methods with higher KNN test accuracy demonstrated better OOD detection performance.
> >
> > Regarding the integration of Barlow Twins loss with mixup regularization, it is worth noting that Barlow Twins can exhibit a tendency to overfit the training data, potentially memorizing specific instances and overemphasizing certain feature representations. This limitation can hinder generalization to downstream tasks, as noted in prior studies [1,3]. Unlike contrastive learning methods such as SimCLR, this challenge impacts the robustness of Barlow Twins' learned features. To mitigate this, the integration of mixup regularization provides a mechanism to enhance generalization by promoting smoother feature representations, reducing overfitting, and improving robustness. We encourage readers to refer to studies such as [1] for additional ablation experiments supporting this approach.
> >
> > We hope this clarification provides a more comprehensive understanding of our decisions and methodology.
> >
> > **Table 1** :: Performance comparison of different labeling methods on MNIST for OOD detection and classification. Metrics include FPR95 (lower is better), AUROC (higher is better), test accuracy for classification on MNIST, and k-NN test accuracy. Ground-truth labels are assumed to achieve 100\% KNN accuracy, while random assignment lacks a computable k-NN accuracy. The OOD dataset used for evaluation is FashionMNIST
> >
> > | Method                | FPR95 (↓) | AUROC (↑) | Test Accuracy (%) | KNN Test Accuracy (%) |
> > |-----------------------|-----------|-----------|-------------------|-----------------------|
> > | **Random Assignment**  | 46.56     | 84.25     | 10.72             | -                     |
> > | **Raw Image Clustering** | 31.87    | 93.22     | 94.17             | 97.68%                |
> > | **GT Labels**          | 10.18     | 97.38     | 99.49             | 100%                  |

---

> > > ### Author Response · Authors · 2024-11-24
> > > **We appreciate the reviewer’s(cCFA) insightful comments. Below, we address the questions raised by the reviewer(Part 3)**
> > >
> > > **Table 2**: We evaluated the OOD detection performance of pseudo-labels generated using various SSL. Our results are shown at the bottom
> > > | Method                | SVHN FPR ↓ | SVHN AUR ↑ | Textures FPR ↓ | Textures AUR ↑ | LSUN-C FPR ↓ | LSUN-C AUR ↑ | LSUN-R FPR ↓ | LSUN-R AUR ↑ | iSUN FPR ↓ | iSUN AUR ↑ | Places365 FPR ↓ | Places365 AUR ↑ | Avg. FPR ↓ | Avg. AUR ↑ | Acc |
> > > |-----------------------|------------|------------|----------------|----------------|--------------|--------------|--------------|--------------|------------|------------|-----------------|-----------------|------------|------------|-----|
> > > | SIMCLR [5]               | 32.68      | 90.75      | 29.33          | 91.35          | 5.67         | 97.29        | 37.15        | 93.92        | 36.40      | 92.62      | 38.72           | 91.87           | 29.99      | 92.96      | 87% |
> > > | BYOL [7]                 | 26.21      | 93.06      | 26.38          | 92.28          | 6.87         | 97.58        | 24.18        | 95.81        | 35.75      | 92.78      | 43.54           | 90.50           | 27.15      | 93.67      | 88% |
> > > | W-MSE [6]                | 28.37      | 92.24      | 26.66          | 92.62          | 4.91         | 97.11        | 27.40        | 93.49        | 36.75      | 92.71      | 40.60           | 90.93           | 27.45      | 93.18      | 88% |
> > >
> > > | Ours (mBT)  | **22.82**      | **93.91**      | **25.18**          | **94.25**          | **1.84**         | **99.61**        | **32.47**        | **93.98**        | **6.71**       | **98.55**      | **32.38**           | **92.63**           | **20.23**      | **95.4**       | **91%** |
> > >
> > >
> > >
> > > ---
> > >
> > > #### Q4:
> > > Finally, the paper lacks the information about reproducibility. Not enough information regarding the algorithms and source codes are provided.
> > >
> > > #### A4:
> > > Our method consists of two stages: SSL training and classifier training, both conducted on the ID dataset. During the SSL stage, we utilized the Adam optimizer along with a cosine annealing learning rate scheduler and a linear warm-up period. We set the weight decay to 1e-6 for 2,000 iterations. In the classifier training stage, we initialized the network with the SSL-trained encoder and employed a Stochastic Gradient Descent (SGD) optimizer. We used a batch size of 128, a momentum of 0.9, and a weight decay of 0.01. For accuracy evaluation and model selection, we randomly selected 5\% of the ID training set. Following the methodology outlined in Yu et al. (2023), we identified the block with the highest norm ratio to compute the out-of-distribution (OOD) score. Please feel free to reach out if you have any specific questions. Thank you.
> > >
> > > Additionally, we have submitted our source code as supplementary material, and all the details mentioned above are available within the code.

---

> > > > ### Author Response · Authors · 2024-11-24
> > > > **We appreciate the reviewer’s(cCFA) insightful comments. Below, we address the questions raised by the reviewer(Part 4)**
> > > >
> > > > ### Questions
> > > > #### Q1:
> > > >  What is cluster-based labels? It should be first described as the reader will not have knowledge about it in prior while reading the paper.
> > > >
> > > > #### A1:
> > > > Thank you for pointing this out. Cluster-based labels refer to pseudo-labels generated by grouping data points based on their feature similarities. In our approach, we utilize the k-means clustering algorithm to categorize data and assign a cluster index (i.e., pseudo-label) to each sample. These pseudo-labels help organize the data in an unsupervised manner, allowing the model to learn from the inherent structure of the data without relying on ground truth labels. We will ensure that a detailed explanation of cluster-based labels is included in the paper for clarity.
> > > >
> > > > ---
> > > >
> > > > #### Q2:
> > > > In Table 2, for random label assignment, why is the AUROC significantly higher than its test accuracy counterpart? While for raw image clustering and GT labels, the difference between AUROC and test accuracy is not that significant?
> > > >
> > > > #### A2:
> > > > In Table 2, AUROC and FPR95 measure OOD detection performance, reflecting the model's ability to differentiate between ID and OOD samples. Test accuracy evaluates the model's classification ability solely on the ID test dataset. For the first row, where random label assignment is used, the model's performance on ID test data is naturally poor, as the labels are arbitrarily assigned and lack meaningful semantic structure. However, the AUROC metric remains relatively high because OOD detection is evaluated independently of the ID classification accuracy.
> > > >
> > > > Regarding the comparison between raw image clustering and GT labels, the apparent difference lies in the tasks being evaluated. Additionally, as highlighted in prior research, "The Curious Case of the Test Set AUROC", AUROC can often present overly optimistic values, which may result in seemingly inflated scores that are not directly comparable to test accuracy.
> > > >
> > > > We will clarify this issue in the revision. Thank you.
> > > >
> > > > ---
> > > >
> > > > #### Q3:
> > > > In line 183, findings and analysis, what does "certain methodology" mean? There should be a clear explanation of this method.
> > > >
> > > > #### A3:
> > > > In this context, "certain methodology" refers to a more logical process than random assignment used to generate pseudo-labels. This involves applying clustering methods to group data samples based on similarities in their raw pixel values, such as color or intensity, without considering high-level semantic features.
> > > >
> > > > ---
> > > > #### Q4:
> > > > How is the method reliant on low-level pixel colors? There is not enough evidence to support this statement. Also, not enough prior information is provided on this matter.
> > > >
> > > > #### A4:
> > > > In Section 3.2, we explore three different methods for generating pseudo-labels. One of these methods is raw data clustering, which relies on low-level pixel colors. In this approach, each image is viewed as a high-dimensional data point, with its dimensionality determined by the image resolution and the number of color channels in each pixel. Images with similar pixel colors are grouped together and assigned the same pseudo-label. However, since each pixel is evaluated independently, without considering the spatial or structural relationships within the image, even minor shifts or transformations can significantly affect the clustering results (i.e., pseudo-labels), leading to inconsistencies.
> > > >
> > > > In contrast to models like ResNet18, which extract hierarchical features through convolutional operations to capture semantic patterns, clustering based on pixel data lacks robustness and generalization. This reliance on low-level attributes restricts the method's ability to effectively distinguish between in-distribution and out-of-distribution samples, as it does not utilize the richer, abstract representations employed in modern classification networks.
> > > >
> > > > #### Q5:
> > > > Based on the pilot study, it is not concluded that pseudo labels need to be derived from high-level representations. Instead, it is concluded that, because existing classification networks use the representations of input from a layer, it performs better. It’s not that, if pseudo labels are derived from these representations, the classification accuracy will increase. This explanation is naive and enough proof is not provided to support this hypothesis. There is not enough proof provided using the pseudo labels and the high-label representations.
> > > >
> > > > #### A5:
> > > > Please refer to A2 in our response regarding our weaknesses. Thank you.

---

> > > > > ### Author Response · Authors · 2024-11-24
> > > > > **We appreciate the reviewer’s(cCFA) insightful comments. Below, we address the questions raised by the reviewer(Part 5)**
> > > > >
> > > > > ---
> > > > > #### Q6:
> > > > > In Section 3.3, it is said that \( Y^A \) and \( Y^B \) are passed through encoder and a projector head, but there is no representation of \( Y^A \) and \( Y^B \) before passing into encoder as images. What is a projector? Is it a decoder?
> > > > >
> > > > > #### A6:
> > > > > \( Y^A \) and \( Y^B \) are two distinct augmentations of a batch of images *X* (line 201 of our submission). These augmentations are processed through an encoder network to extract feature representations (as shown in Figure 1(a), and line 203 of our submission). These representations are then passed through a projector head, which maps them to a lower-dimensional space.
> > > > >
> > > > > In self-supervised learning (SSL), a projector usually refers to a multilayer perceptron (MLP) that transforms the learned representations from the backbone into a latent space for optimizing SSL objectives, such as contrastive loss. Unlike a decoder, which reconstructs data or specific attributes (for example, pixels in autoencoders), the projector is primarily focused on refining feature representations to improve alignment with the SSL task.
> > > > >
> > > > > #### Q7:
> > > > > What is \( Y^M \) here and where does it come from?
> > > > > #### A7:
> > > > > \( Y^M \) is a batch of images obtained by applying the MixUp augmentation [4]. This augmentation blends two distinct augmentations of a batch of images *X*, \( Y^A \) and a randomly shuffled version of \( Y^B \).
> > > > >
> > > > > \( Y^M \) is generated through a linear interpolation of \( Y^A \) and \( Y^B \). For additional details regarding \( Y^M \), please refer to the lines (203 to 205).
> > > > >
> > > > > #### Q8:
> > > > > The schematics of the figure 1 is quite irrelevant to the explanation provided in section 3.3. For instance, what does dotted vertical line represent between two encoders. And Where is \( Y^M \)  coming from ?
> > > > >
> > > > > #### A8:
> > > > > The dotted vertical lines in Figure 1 indicate that the same network, utilizing shared weights (specifically, the encoder \( f_e \) and projector \( f_p \) are applied to process different inputs (\( Y^A \), \( Y^B \), and \( Y^M \)). This weight-sharing ensures that all three inputs are transformed using identical parameters, which reinforces the concept of consistency in representation learning across augmented views. This shared architecture is a fundamental aspect of the Barlow Twins method, which relies on contrasting representations of different views of the same data point within a unified network setup. We will update the figure's details to enhance clarity for the readers.
> > > > >
> > > > > Regarding \( Y^M \), it represents a batch of images created by linearly interpolating the batch images \( Y^A \) and \( Y^B' \), where \( Y^B' \) is a shuffled version of \( Y^B \). Please also refer to line 204 in our submission. Thank you.
> > > > >
> > > > >
> > > > >
> > > > > **References:**
> > > > > 1. Wele Gedara Chaminda Bandara, Celso M De Melo, and Vishal M Patel. *Guarding Barlow Twins against Overfitting with Mixed Samples*. arXiv e-prints, pp. arXiv–2312, 2023.
> > > > > 2. Shiyu Liang, Yixuan Li, and Rayadurgam Srikant. *Enhancing the Reliability of Out-of-Distribution Image Detection in Neural Networks*. arXiv preprint arXiv:1706.02690, 2017.
> > > > > 3. Haoqing Wang, Xun Guo, Zhi-Hong Deng, and Yan Lu. *Rethinking Minimal Sufficient Representation in Contrastive Learning*. In Proceedings of the IEEE/CVF Conference on Computer Vision and Pattern Recognition, pp. 16041–16050, 2022.
> > > > > 4. Hongyi Zhang, Moustapha Cisse, Yann N Dauphin, and David Lopez-Paz. *Mixup: Beyond Empirical Risk Minimization*. arXiv preprint arXiv:1710.09412, 2017.
> > > > > 5. Ting Chen, Simon Kornblith, Mohammad Norouzi, and Geoffrey Hinton. *A simple framework for contrastive learning of visual representations*. In *International Conference on Machine Learning*, pp. 1597–1607. PMLR, 2020.
> > > > >
> > > > > 6. Aleksandr Ermolov, Aliaksandr Siarohin, Enver Sangineto, and Nicu Sebe. *Whitening for self-supervised representation learning*. In *International Conference on Machine Learning*, pp. 3015–3024. PMLR, 2021.
> > > > >
> > > > > 7. Jean-Bastien Grill, Florian Strub, Florent Altché, Corentin Tallec, Pierre Richemond, Elena Buchatskaya, Carl Doersch, Bernardo Avila Pires, Zhaohan Guo, Mohammad Gheshlaghi Azar, et al. *Bootstrap your own latent: A new approach to self-supervised learning*. In *Advances in Neural Information Processing Systems*, 33:21271–21284, 2020.

---

> > > > > > ### Comment · Reviewer_cCFA · 2024-11-25
> > > > > >
> > > > > > Response to A7 and A8: Although the explanation is present in the manuscript about the inputs, It is recommended for the authors to clearly represent them in the Fig. 1 (such as adding a mixup block between Y^A and Y^B to obtain Y^M) and present them according to the explaination. Regarding, A8, it is suggested the explanation be included clearly in the description of the figure.

---

> > > > > > > ### Author Response · Authors · 2024-11-29
> > > > > > >
> > > > > > > Dear reviewer,
> > > > > > >
> > > > > > > We have updated Figure 1 in our revised manuscript. Thank you very much for the suggestion.
> > > > > > >
> > > > > > > The authors.

---

> > > > > ### Comment · Reviewer_cCFA · 2024-11-25
> > > > >
> > > > > Thank you for trying to clarify all the weaknesses and questionaries. However, there are still few minor issues with the writings, where a thorough proofread should be done.

---

> > > > ### Comment · Reviewer_cCFA · 2024-11-25
> > > >
> > > > Thank you for the clarification of the comment.

---

> > > ### Comment · Reviewer_cCFA · 2024-11-25
> > >
> > > Thank you for the clarification for this question. It is highly appreciated that the authors have provided additional evidence to support the use of BT loss in their method. Compared to other counterparts, mBT method achieves superior performance across all OOD detection metrics for all datasets.

---

> > ### Comment · Reviewer_cCFA · 2024-11-25
> >
> > Thank you for the justification of the question. It is highly appreciated that the authors have updated the submission based on the question and clarifying the comments.

---

### Official Review · Reviewer_chkx · 2024-11-03

**Soundness:** 3
**Presentation:** 4
**Contribution:** 3
**Rating:** 6
**Confidence:** 3

**Summary:**

This paper seeks to bridge the gap between supervised and unsupervised OOD detection by proposing a new approach that trains models to categorize data into pseudo-classes. Through self-supervised learning (SSL), raw data is transformed into representations, which are then clustered to create pseudo-labels. These pseudo-labels are used to train a classifier capable of OOD detection. The method is evaluated against several supervised and unsupervised OOD detection techniques, and an in-depth discussion of its various aspects is provided.

**Strengths:**

1. The writing is straightforward and clear, and the relevant works in introduction and related works part are also clear. The logic of the method section is coherent, and the experimental setup is thoroughly described.
2. This paper conducts extensive experiments to validate the effectiveness of the proposed approach across various supervised and unsupervised OOD detection methods, as well as ID and OOD datasets and model architectures.
3. The experiments presented in the discussion are engaging and provide meaningful insights.

**Weaknesses:**

1. The novelty of this work is somewhat limited, as SSL and pseudo-labeling have been previously introduced in the OOD detection domain [1], albeit with a different approach. This reduces the contribution’s novelty primarily to the method of generating pseudo-labels. Additionally, [1] should be included in your experiments given the strong similarities to your work.
2. Comparison to supervised methods – the authors demonstrate that their method outperforms supervised methods, but they do not clarify the experimental settings used for this comparison (e.g., how do they compare ID performance). See further questions below.
3. Computational effort comparison – training the SSL network introduces additional computational requirements, which were not addressed in comparison to other unsupervised approaches. Furthermore, the networks being compared are not of the same size. The authors should, at minimum, include a column in the table indicating the network sizes to ensure a fair comparison
4. Experiment statistical significance – the reported results are not averaged across multiple seeds, which is crucial in the OOD domain, especially when results are close (e.g., SVHN as ID in Table 3).


Minor comments:

5. The method section would benefit from a formulaic explanation of the OOD detection decision function (often related to as $G(x,\tau)$), including details on how the OOD cut-off threshold is determined.

6. Adding the names of the OOD detection methods to the tables would improve readability.

[1] Mohseni, S., Pitale, M., Yadawa, J. B. S., & Wang, Z. (2020, April). Self-supervised learning for generalizable out-of-distribution detection. In Proceedings of the AAAI conference on artificial intelligence (Vol. 34, No. 04, pp. 5216-5223).‏

**Questions:**

1. Learned feature space comparison – to confirm that the SSL embeddings are central to the method’s success, an experiment comparing them to the embedding spaces of off-the-shelf networks is needed. For example, the feature spaces of ResNet or CLIP could be used for comparison. This would also partially address weakness 3 above.
2. It would be interesting to assess whether pseudo labels are even necessary by automatically labeling the ID datasets using state-of-the-art open-set classification approaches like Grounded SAM or Grounding DINO [2].
3. In the supervised methods experiment (Section 4.2), what are the ID accuracy results for the different backbones? Additionally, what pseudo-labels are used in this scenario? If ground-truth labels are employed, then the approach effectively reduces to the vanilla FeatureNorm (Yu et al., 2023). Alternatively, if only OOD detection performance is measured, the task becomes a classic anomaly detection problem.
4. Is there a specific reason for choosing FeatureNorm (Yu et al., 2023) over other state-of-the-art OOD detection methods (e.g., Mahalanobis or Energy)? Since you train a standard classification network in the final step, it would be beneficial to evaluate how other OOD detection methods perform with the trained network.
5. Is there a reason why FPR results were not shown in Table 3?

[2] Liu, S., Zeng, Z., Ren, T., Li, F., Zhang, H., Yang, J., ... & Zhang, L. (2023). Grounding dino: Marrying dino with grounded pre-training for open-set object detection. arXiv preprint arXiv:2303.05499.‏

---

> ### Author Response · Authors · 2024-11-22
> **We appreciate the reviewer’s(chkx) insightful comments. Below, we address the questions raised by the reviewer(Part 1)**
>
> ### Weaknesses
>
> #### Q1:
> The novelty of this work is somewhat limited, as SSL and pseudo-labeling have been previously introduced in the OOD detection domain Mohseniet al. [9], albeit with a different approach. This reduces the contribution’s novelty primarily to the method of generating pseudo-labels. Additionally, Mohseniet al. [9] should be included in your experiments, given the strong similarities to your work.
>
> #### A1:
> We appreciate your feedback. However, we contend that our insight - that supervised and unsupervised OOD detection methods are fundamentally similar - is indeed innovative. Specifically, they utilize ID data to learn a particular task. Supervised methods learn to classify ID data, while unsupervised methods learn to reconstruct it. These methods then rely on the performance discrepancy between ID and OOD data to identify OOD samples. Although simple, this approach is far from intuitive; if it were, it would be challenging to justify the longstanding bifurcation of research into supervised and unsupervised OOD detection pathways. The potential publication of our manuscript could inspire a merging of these research paths, marking a significant shift in the field of OOD detection methods in the future.
>
> We believe that novelty should not be confined to the development of base algorithms or training techniques alone. Our insight contributes at a conceptual level, where utilizing pseudo-labels allows us to train supervised methods without manual annotation, achieving higher detection accuracy and faster detection speeds. While pursuing algorithmic novelties along the two traditional paths might appear novel, it often lacks the practical benefits that our approach offers by combining the strengths of both supervised and unsupervised methodologies.
>
> While we recognize the foundational work by Mohseniet al.[9] in the OOD detection domain, our approach differs significantly in both structure and conceptual focus. Mohseniet al.[9] employs a two-step process, beginning with fully supervised training, which necessitates manual annotation for ID data, followed by SSL applied to unlabeled OOD data. In contrast, our approach requires only unlabeled ID data for training. Therefore, comparing these two methods is misleading because Mohseniet al.[9] additionally utilizes labels from ID data and even unlabeled OOD data during training.
>
>
> ---
>
> #### Q2:
> Comparison to supervised methods – the authors demonstrate that their method outperforms supervised methods, but they do not clarify the experimental settings used for this comparison (e.g., how do they compare ID performance).
>
> #### A2:
> We followed the methodology detailed in Yu et al.[4] to ensure a fair and consistent comparison. As described in lines 316-316 of our manuscript, the ID and OOD datasets were configured such that both the SSL model and the OOD detector were trained exclusively on the ID training dataset. For evaluation, we combined the ID testing dataset with the OOD dataset to measure the performance of the OOD detection methods. We hope this clarifies the experimental settings used in our comparisons. Please let us know if additional details would further address your concerns.
>
> ---

---

> > ### Author Response · Authors · 2024-11-22
> > **We appreciate the reviewer’s(chkx) insightful comments. Below, we address the questions raised by the reviewer(Part 2)**
> >
> > #### Q3:
> > Computational effort comparison – training the SSL network introduces additional computational requirements, which were not addressed in comparison to other unsupervised approaches. Furthermore, the networks being compared are not of the same size. The authors should, at minimum, include a column in the table indicating the network sizes to ensure a fair comparison.
> >
> > #### A3:
> > Thank you for your thoughtful comments regarding the computational effort and network size comparison. We acknowledge that training the SSL network introduces additional computational requirements; however, in the context of OOD detection, the training time is typically considered less critical because the model is trained only once. The focus of most OOD detection studies, including ours, is on the inference stage, where response time and detection accuracy are the primary metrics of interest. This is analogous to large language models (LLMs), where despite extensive training times, users primarily care about their response times during deployment.
> >
> > Regarding the comparison of network sizes, we note that unsupervised methods are often built on fundamentally different architectures based on their conceptual frameworks. For example, earlier approaches used encoder-decoder architectures that could reconstruct an input image in a single pass, while later methods, such as those based on diffusion models, require numerous denoising processes to reconstruct a single image. As a result, it is uncommon in the literature to compare network sizes directly, as they are not indicative of practical inference performance or detection accuracy.
> >
> > Instead, the focus is generally on the time required to process each test sample and the corresponding detection accuracy. In our study, as shown in Table 1, the state-of-the-art unsupervised OOD detection method based on DDPM requires 28 seconds to batch of 16 images. In contrast, our method, like FeatureNorm, requires only 0.025 seconds, providing a significant advantage in inference speed.
> >
> > We hope this clarifies our approach and the rationale behind the metrics chosen for comparison. Thank you again for your valuable feedback.

---

> > > ### Author Response · Authors · 2024-11-22
> > > **We appreciate the reviewer’s(chkx) insightful comments. Below, we address the questions raised by the reviewer(Part 3)**
> > >
> > > #### Q4:
> > > Experiment statistical significance – the reported results are not averaged across multiple seeds, which is crucial in the OOD domain, especially when results are close (e.g., SVHN as ID in Table 3).
> > >
> > > #### A4:
> > > Thank you for pointing this out. To address the concern, we have provided the averaged results of our method across three different seeds in the updated Table 1 and Table 2. As observed, the results were consistent and stable, reinforcing the reliability of our method.
> > >
> > >
> > >
> > > **Table 1**: The averaged OOD detection results were calculated using three different seeds, along with their standard deviation..
> > > |           | **FMnist**          |     **FMnist**        |    **FMnist**         | **CIF10**     |       **CIF10**     |     **CIF10**       |     **CIF10**       | **CelebA**         |  **CelebA**          |   **CelebA**         |     **CelebA**       | **SVHN**          |      **SVHN**      |     **SVHN**       |       **SVHN**     |
> > > |-----------|---------------------|-----------|-----------|------------------|-----------|-----------|-----------|--------------------|-----------|-----------|-----------|-------------------|-----------|-----------|-----------|
> > > |           | Mnist      | V         | H         | Svhn             | CelA      | V         | H         | Cif10             | Svhn      | V         | H         | Cif10             | CelA      | V         | H         |
> > > | **Our**   | 96.47  &nbsp;± 0.25       | 86.97 ±0.15 | 70.27 ±0.21 | 99.13 ±0.06    | 76.73 ±0.38 | 70.03 ±0.38 | 49.97 ±0.06 | 99.77 ±0.06    | 100.00 ±0.00 | 98.57 ±0.06 | 47.43 ±2.74 | 99.47 ±0.06    | 99.00 ±0.01 | 66.17 ±0.95 | 65.90 ±0.72 |
> > >
> > > ---
> > >
> > > **Table 2**: The averaged OOD detection results were calculated using three different seeds, along with their standard deviation.
> > >
> > > | Method               | **SVHN** FPR↓ | **SVHN** AUR↑ | **Textures** FPR↓ | **Textures** AUR↑ | **LSUN-C** FPR↓ | **LSUN-C** AUR↑ | **LSUN-R** FPR↓ | **LSUN-R** AUR↑ | **iSUN** FPR↓ | **iSUN** AUR↑ | **Places365** FPR↓ | **Places365** AUR↑ | **Average** FPR↓ | **Average** AUR↑ |
> > > |----------------------|---------------|---------------|-------------------|-------------------|-----------------|-----------------|-----------------|-----------------|---------------|---------------|---------------------|-------------------|------------------|-------------------|
> > > | Res18_CIFAR10         | 21.84 ± 2.10  | 93.86 ± 0.16  | 23.85 ± 1.27      | 94.53 ± 0.42      | 1.88 ± 0.04     | 99.53 ± 0.15    | 35.52 ± 2.95    | 93.61 ± 0.35    | 6.96 ± 1.81    | 98.20 ± 0.73    | 33.76 ± 1.21      | 92.30 ± 0.29      | 20.63 ± 0.36    | 95.31 ± 0.10     |
> > > | WRN28_CIFAR10         | 4.85 ± 0.52   | 99.11 ± 0.12  | 13.35 ± 1.60      | 99.13 ± 0.09      | 1.87 ± 0.15     | 99.49 ± 0.11    | 18.30 ± 0.83    | 96.66 ± 0.03    | 13.06 ± 1.72   | 97.26 ± 0.77    | 25.23 ± 3.10      | 94.80 ± 0.32      | 12.78 ± 0.53    | 97.41 ± 0.14     |
> > > | VGG11_CIFAR10         | 27.16 ± 0.21  | 93.37 ± 0.22  | 26.92 ± 0.53      | 92.12 ± 0.22      | 2.04 ± 0.62     | 99.52 ± 0.11    | 74.97 ± 1.14    | 85.07 ± 0.09    | 16.10 ± 0.18   | 96.85 ± 0.07    | 62.30 ± 0.57      | 85.14 ± 0.46      | 35.53 ± 0.77    | 91.92 ± 0.08     |
> > > | Res18_CIFAR100        | 54.36 ± 0.64  | 89.00 ± 0.58  | 63.52 ± 0.37      | 82.67 ± 0.15      | 7.88 ± 0.42     | 98.35 ± 0.14    | 77.65 ± 0.24    | 77.75 ± 0.16    | 57.54 ± 1.30   | 87.20 ± 0.72    | 67.92 ± 0.78      | 82.80 ± 0.53      | 54.84 ± 0.14    | 86.26 ± 0.07     |
> > >
> > >
> > > ### Minor Comments
> > > #### Q5:
> > > The method section would benefit from a formulaic explanation of the OOD detection decision function, including details on how the OOD cut-off threshold is determined.
> > >
> > > #### A5 :
> > > Thank you for your suggestion. The cut-off threshold is set to ensure that 95\% of ID data is accurately classified as ID, resulting in a true positive rate of 95\%.
> > >
> > >
> > > #### Q6 :
> > > Adding the names of the OOD detection methods to the tables would improve readability.
> > > #### A6 :
> > > We will add the names of the OOD detection methods in the revision. Thank you.

---

> > > > ### Author Response · Authors · 2024-11-22
> > > > **We appreciate the reviewer’s(chkx) insightful comments. Below, we address the questions raised by the reviewer(Part 4)**
> > > >
> > > > ### Questions
> > > >
> > > > #### Q1:
> > > > Learned feature space comparison – to confirm that the SSL embeddings are central to the method’s success, an experiment comparing them to the embedding spaces of off-the-shelf networks is needed. For example, the feature spaces of ResNet or CLIP could be used for comparison. This would also partially address weakness 3 above.
> > > >
> > > > #### A1:
> > > > Thank you for this thoughtful suggestion. In our study, we observed that the coherence of pseudo-labels between training and testing ID datasets plays a critical role in achieving better OOD detection performance. While the learned feature space significantly influences this coherence, the quality of pseudo-labels is also affected by the data embeddings within the same feature space. In other words, the pseudo-label quality depends not only on the network backbone but also on the choice of the SSL method.
> > > >
> > > > To date, one commonly used metric for evaluating SSL embeddings is KNN accuracy. In our experiments, we found that KNN accuracy strongly correlates with OOD detection performance (as shown in Table 6 of our paper). Higher KNN accuracy indicates better pseudo-label quality, which, in turn, leads to improved OOD detection performance. This suggests that selecting a combination of off-the-shelf networks and SSL methods with higher KNN accuracy can produce superior pseudo-labels for OOD detection.
> > > >
> > > > To address your suggestion, we conducted additional experiments comparing the pseudo-labels generated using two different backbones, ViT-S/8 and ResNet-50, for the ImageNet dataset. Both backbones were trained using the DINO SSL method, and we used ResNet-50 as the classifier in subsequent OOD detection experiments. Table 3 presents the results of these experiments, showing that pseudo-labels generated with ViT-S/8, which has higher KNN accuracy, outperform those generated with ResNet-50. This aligns with our explanation and further highlights the importance of the learned feature space and its embeddings in the success of our method.
> > > >
> > > > We appreciate your suggestion, which has provided an opportunity to reinforce the connection between SSL embeddings, KNN accuracy, and OOD detection performance.
> > > >
> > > >
> > > > **Table 3**: We assess the OOD detection performance using ImageNet as the ID dataset. The first row lists the OOD datasets. The pseudo labels are generated  through different SSL backbones and trained on Resnet-50 classifier
> > > > | Method      | **INaturalist** FPR↓ | **INaturalist** AUR↑ | **SUN** FPR↓ | **SUN** AUR↑ | **Places** FPR↓ | **Places** AUR↑ | **Texture** FPR↓ | **Texture** AUR↑ | **Average** FPR↓ | **Average** AUR↑ | **KNN Accuracy** |
> > > > |-------------|----------------------|----------------------|--------------|--------------|-----------------|-----------------|------------------|------------------|------------------|------------------|------------------|
> > > > | ResNet-50   | 29.25                | 93.25                | 43.72        | 89.62        | 58.4            | 83.15           | 27.1             | 93.41            | 39.62            | 89.85            | 75.3%            |
> > > > | VIT(S/8)    | **28.56**            | **93.46**            | **42.81**    | **90.93**    | **56.67**       | **83.58**       | **27.25**        | **93.21**        | **38.82**        | **90.35**        | **79.7%**        |
> > > >
> > > >
> > > >
> > > > ---
> > > > #### Q2:
> > > > It would be interesting to assess whether pseudo labels are even necessary by automatically labeling the ID datasets using state-of-the-art open-set classification approaches like Grounded SAM or Grounding DINO.
> > > >
> > > > #### A2:
> > > > Thank you for your excellent suggestion. We believe that using open-set classification approaches, such as Grounded SAM or DINO, to generate pseudo-labels would be advantageous. However, we did not implement this in our study because these models were trained on open datasets. Comparing our method with baseline approaches could be considered unfair in this context.

---

> > > > > ### Author Response · Authors · 2024-11-22
> > > > > **We appreciate the reviewer’s(chkx) insightful comments. Below, we address the questions raised by the reviewer(Part 5)**
> > > > >
> > > > > #### Q3:
> > > > > In the supervised methods experiment (Section 4.2), what are the ID accuracy results for the different backbones? Additionally, what pseudo-labels are used in this scenario? If ground-truth labels are employed, then the approach effectively reduces to the vanilla FeatureNorm[4]. Alternatively, if only OOD detection performance is measured, the task becomes a classic anomaly detection problem.
> > > > >
> > > > >
> > > > > #### A3:
> > > > > - For CIFAR-10:
> > > > >   - ResNet18: 93%
> > > > >   - WRN28_10: 93%
> > > > >   - VGG11: 87%
> > > > >
> > > > > - For CIFAR-100 with ResNet18: 79%.
> > > > >
> > > > > In these experiments, pseudo-labels are generated via clustering, with the datasets partitioned into 20 clusters for CIFAR-10 and 200 clusters for CIFAR-100. These pseudo-labels are derived entirely from the dataset without relying on ground-truth labels, thereby preserving the unsupervised nature of our approach. It is important to emphasize that our method does not utilize ground-truth labels, and thus, it does not reduce to FeatureNorm [4]. The use of clustering-derived pseudo-labels ensures the method remains distinct and unsupervised. Additionally, it is worth noting that unsupervised OOD detection methods are inherently not designed for ID dataset classification.
> > > > >
> > > > > Furthermore, many supervised OOD detection methods trained using ground-truth labels, such as LogitNorm[11] and MaxLogit[3], often experience degraded classification ability for ID datasets due to their optimization mechanisms being specifically tailored for OOD detection. This further underscores the advantages of our approach, which avoids such trade-offs by using pseudo-labels.

---

> > > > > > ### Author Response · Authors · 2024-11-22
> > > > > > **We appreciate the reviewer’s(chkx) insightful comments. Below, we address the questions raised by the reviewer(Part 6)**
> > > > > >
> > > > > > #### Q4:
> > > > > > Is there a specific reason for choosing FeatureNorm (Yu et al., 2023) over other state-of-the-art OOD detection methods (e.g., Mahalanobis or Energy)? Since you train a standard classification network in the final step, it would be beneficial to evaluate how other OOD detection methods perform with the trained network.
> > > > > >
> > > > > > #### A4:
> > > > > > We chose FeatureNorm [4] as the primary state-of-the-art supervised OOD detection method in our study for the following reasons:
> > > > > > 1. **Superior OOD Detection Performance:** FeatureNorm consistently demonstrates higher OOD detection performance compared to methods based on Mahalanobis distance or Energy scores.
> > > > > >
> > > > > > 2. **Reproducibility:** During our reproduction of results from baseline methods, we observed that many methods exhibited significant performance fluctuations in OOD detection across different training epochs. This variability is likely due to the inherent differences between the ID and OOD datasets. In contrast, FeatureNorm showed minimal sensitivity to such fluctuations, allowing us to reliably reproduce its reported results, which further reinforced our decision to use it in our experiments.
> > > > > >
> > > > > > 3. **Theoretical Foundations:** Park et al. (2023)[10] provided a theoretical explanation for the effectiveness of feature norms in detecting OOD samples. This solid theoretical grounding supported our choice to pair FeatureNorm with pseudo-labels for OOD detection.
> > > > > >
> > > > > > To address your suggestion, we conducted additional experiments by applying other supervised OOD detection methods in combination with our pseudo-labels. The results, presented in Table 4, indicate that these methods underperformed compared to FeatureNorm in terms of OOD detection accuracy. We believe this further validates the suitability of FeatureNorm as a benchmark in our framework.
> > > > > >
> > > > > >
> > > > > > **Table 4**: We extended several baseline supervised OOD detection methods by incorporating pseudo-labels determined through self-supervised learning. We conducted experiments in accordance with the benchmark settings established in our paper. Our approach of combining FeatureNorm [4] with pseudo-labels yielded the best performance..
> > > > > >
> > > > > > | Method                 | **SVHN** FPR↓ | **SVHN** AUR↑ | **Textures** FPR↓ | **Textures** AUR↑ | **LSUN-C** FPR↓ | **LSUN-C** AUR↑ | **LSUN-R** FPR↓ | **LSUN-R** AUR↑ | **iSUN** FPR↓ | **iSUN** AUR↑ | **Places365** FPR↓ | **Places365** AUR↑ | **Average** FPR↓ | **Average** AUR↑ |
> > > > > > |------------------------|---------------|---------------|-------------------|-------------------|-----------------|-----------------|-----------------|-----------------|---------------|---------------|---------------------|-------------------|-------------------|-------------------|
> > > > > > | Energy [8]  | 33.43         | 92.98         | 42.66             | 90.51             | 18.98           | 97.35           | 28.33           | 93.29           | 28.18         | 93.31         | 54.22               | 88.19             | 34.3              | 92.60             |
> > > > > > | Mahalanobis [7] | 28.39         | 93.98         | 23.56             | 94.2              | 4.04            | 99.19           | 38.89           | 92.64           | 16.08         | 96.77         | 48.45               | 91.44             | 26.57             | 94.70             |
> > > > > > | MSP [2] | 17.94         | 88.53         | 32.91             | 89.87             | 18.65           | 89.36           | 88.24           | 61.83           | 46.04         | 88.03         | 54.48               | 85.62             | 43.3              | 83.87             |
> > > > > > | DML+ [53]   | 28.39         | 93.98         | 23.56             | 94.2              | 4.04            | 99.19           | 38.89           | 92.64           | 16.08         | 96.77         | 48.45               | 91.44             | 26.57             | 94.70             |
> > > > > > | NAN [10]     | 35.62         | 91.47         | 75.5              | 86.25             | 2.81            | 99.37           | 40.15           | 99.28           | 16.60         | 95.75         | 35.66               | 92.17             | 34.6              | 93.10             |
> > > > > > | Ours (FeatureNorm)         | **22.82**     | **93.91**     | **25.18**         | **94.25**         | **1.84**        | **99.61**       | **32.47**        | **93.98**       | **6.71**       | **98.55**       | **32.38**           | **92.63**         | **20.23**         | **95.4**          |

---

> > > > > > > ### Author Response · Authors · 2024-11-22
> > > > > > > **We appreciate the reviewer’s(chkx) insightful comments. Below, we address the questions raised by the reviewer(Part 7)**
> > > > > > >
> > > > > > > #### Q5:
> > > > > > > Is there a reason why FPR results were not shown in Table 3?
> > > > > > > #### A5:
> > > > > > > Thank you for pointing this out. In Table 3 of the manuscript, we directly excerpted the AUROC values from the original works of [1] and [4]. Both of these papers exclusively reported AUROC scores as the primary metric for evaluating method performance, and they did not include FPR results in their evaluations.
> > > > > > >
> > > > > > > Additionally, when we reproduced the results using their released codes, we found that the AUROC scores we obtained did not fully align with the values reported in their papers. To avoid potential discrepancies and ensure consistency, we chose to adhere to the evaluation setup in [1] and excluded FPR results in Table 3. We appreciate your understanding on this matter and are open to further suggestions if additional metrics would enhance the clarity of our comparisons.
> > > > > > >
> > > > > > >
> > > > > > >
> > > > > > >
> > > > > > >
> > > > > > >
> > > > > > >
> > > > > > >
> > > > > > > **References:**
> > > > > > > 1. Mark S. Graham, Walter H. L. Pinaya, Petru-Daniel Tudosiu, Parashkev Nachev, Sebastien Ourselin, and Jorge Cardoso. *Denoising diffusion models for out-of-distribution detection*. In Proceedings of the IEEE/CVF Conference on Computer Vision and Pattern Recognition, pp. 2947–2956, 2023.
> > > > > > >
> > > > > > > 2. Dan Hendrycks and Kevin Gimpel. *A baseline for detecting misclassified and out-of-distribution examples in neural networks*. arXiv preprint arXiv:1610.02136, 2016.
> > > > > > >
> > > > > > > 3. Dan Hendrycks, Steven Basart, Mantas Mazeika, Andy Zou, Joe Kwon, Mohammadreza Mostajabi, Jacob Steinhardt, and Dawn Song. *Scaling out-of-distribution detection for real-world settings*. arXiv preprint arXiv:1911.11132, 2019.
> > > > > > >
> > > > > > > 4. Yeonguk Yu, Sungho Shin, Seongju Lee, Changhyun Jun, and Kyoobin Lee. *Block selection method for using feature norm in out-of-distribution detection*. In Proceedings of the IEEE/CVF Conference on Computer Vision and Pattern Recognition, pp. 15701–15711, 2023.
> > > > > > >
> > > > > > > 5. Zihan Zhang and Xiang Xiang. *Decoupling maxlogit for out-of-distribution detection*. In Proceedings of the IEEE/CVF Conference on Computer Vision and Pattern Recognition, pp. 3388–3397, 2023.
> > > > > > >
> > > > > > > 6. Hamidreza Kamkari, Brendan Leigh Ross, Jesse C. Cresswell, Anthony L. Caterini, Rahul G. Krishnan, and Gabriel Loaiza-Ganem. *A geometric explanation of the likelihood OOD detection paradox*. arXiv preprint arXiv:2403.18910, 2024.
> > > > > > >
> > > > > > > 7. Kimin Lee, Kibok Lee, Honglak Lee, and Jinwoo Shin. *A simple unified framework for detecting out-of-distribution samples and adversarial attacks*. Advances in Neural Information Processing Systems, 31, 2018.
> > > > > > >
> > > > > > > 8. Weitang Liu, Xiaoyun Wang, John Owens, and Yixuan Li. *Energy-based out-of-distribution detection*. Advances in Neural Information Processing Systems, 33:21464–21475, 2020.
> > > > > > >
> > > > > > > 9. Sina Mohseni, Mandar Pitale, J.B.S. Yadawa, and Zhangyang Wang. *Self-supervised learning for generalizable out-of-distribution detection*. In Proceedings of the AAAI Conference on Artificial Intelligence, volume 34, pp. 5216–5223, 2020.
> > > > > > >
> > > > > > > 10. Jaewoo Park, Jacky Chen Long Chai, Jaeho Yoon, and Andrew Beng Jin Teoh. *Understanding the feature norm for out-of-distribution detection*. In Proceedings of the IEEE/CVF International Conference on Computer Vision, pp. 1557–1567, 2023.
> > > > > > >
> > > > > > > 11. Hongxin Wei, Renchunzi Xie, Hao Cheng, Lei Feng, Bo An, and Yixuan Li. *Mitigating neural network overconfidence with logit normalization*. In International Conference on Machine Learning, pp. 23631–23644. PMLR, 2022.

---

### Official Review · Reviewer_S5MM · 2024-11-04

**Soundness:** 3
**Presentation:** 3
**Contribution:** 2
**Rating:** 3
**Confidence:** 4

**Summary:**

This paper proposes to detect out-of-distribution samples with unlabeled training data, by assigning pseudo-labels from self-supervised features clustering. Empirical results demonstrate the superiority of the method over state-of-the-art supervised and unsupervised approaches.

**Strengths:**

1. Detecting out-of-distribution samples with unlabeled training data is meaningful because the detector from supervised learning usually comes with high labeling costs. This paper explores the possibility of an unsupervised OOD detector with comparable or even better performance than the supervised ones.
2. This paper observes that pseudo-labels from self-supervised representation clustering can replace and outperform the ground-truth labels, which is surprising.
3. The benchmarks are comprehensive and presented clearly.

**Weaknesses:**

1. The motivation is not clear. Specifically, what is the meaning of the insight that "the success of supervised methods relies on a classification strategy that may not align with real-world understanding", and what is the objective and function of Section 3.2?
2. The method is neither novel nor solid. The methodology design, both the self-supervised representation learning and detector, depends on existing works. Additionally, the key mechanism behind the superiority of pseudo-labels from SSL is not well understood.
3. The experiment results are not convincing. On the one hand, the unsupervised benchmark seems carefully designed and not commonly used. For example, vertically and horizontally flipped versions of each ID dataset act as extra OOD datasets. On the other hand, the details about tuning of pseudo-categories number is missing. How does the author evaluate the classification accuracy with more categories than ID training datasets?

**Questions:**

1. Given that SSL can extract features with semantic information, is it necessary to train the model SSL pseudo-labels?
2. Why do the pseudo-labels generated by SSL offer advantages over GT labels? The author should give a deeper understanding.

---

> ### Author Response · Authors · 2024-11-21
> **We appreciate the reviewer’s(S5MM) insightful comments. Below, we address the questions raised by the reviewer(Part 1)**
>
> ## Reviewer S5MM
>
> ### Weaknesses
>
> #### Q1:
> The motivation is not clear. Specifically, what is the meaning of the insight that "the success of supervised methods relies on a classification strategy that may not align with real-world understanding," and what is the objective and function of Section 3.2?
>
> #### A1:
> Our conceptual insight is that supervised methods can detect out-of-distribution (OOD) samples because they have learned a classification logic specific to the in-distribution (ID) dataset rather than genuinely recognizing real-world objects (such as cars or people). This classification logic enables the model to generalize this learned logic to testing data. Consequently, the difference in performance between ID and OOD datasets can effectively distinguish the two. Section 3.2 validates that as long as the classification logic within the ID data is coherent, the model can effectively learn this logic, achieving high OOD detection accuracy.
>
> ---
>
> #### Q2:
> The method is neither novel nor solid. The methodology design, both the self-supervised representation learning and detector, depends on existing works.
>
> #### A2:
> We appreciate your feedback. However, we contend that our insight—that supervised and unsupervised OOD detection methods are fundamentally similar—is indeed innovative. Specifically, they utilize ID data to learn a particular task. Supervised methods learn to classify ID data, while unsupervised methods learn to reconstruct it. These methods then rely on the performance discrepancy between ID and OOD data to identify OOD samples.While our method builds on existing works, it represents a conceptual contribution by combining strengths of supervised and unsupervised methods. This could inspire a unified research path for OOD detection in the future.
>
> We believe that novelty should not be confined to the development of base algorithms or training techniques alone. Our insight contributes at a conceptual level, where utilizing pseudo-labels allows us to train supervised methods without manual annotation, achieving higher detection accuracy and faster detection speeds. While pursuing algorithmic novelties along the two traditional paths might appear novel, it often lacks the practical benefits that our approach offers by combining the strengths of both supervised and unsupervised methodologies
>
> ---
>
> #### Q3:
> The experimental results are not convincing. The unsupervised benchmark seems carefully designed and not commonly used. For example, vertically and horizontally flipped versions of each ID dataset act as extra OOD datasets.
>
> #### A3:
>  Thank you very much for your thoughtful feedback. For the unsupervised benchmarking, we followed the exact procedure outlined in prior work by [1,2], without introducing any modifications from our side. The results of the baseline methods reported in our study were directly copied from  [1,2], preserving the integrity of the original benchmarking criteria. We agree with the reviewer that considering vertically and horizontally flipped versions of an ID dataset to be OOD may not be adequate, and we are happy to remove the experiment results in the revision.
>
> ---

---

> > ### Author Response · Authors · 2024-11-21
> > **We appreciate the reviewer’s(S5MM) insightful comments. Below, we address the questions raised by the reviewer(Part 2)**
> >
> > #### Q4:
> > The details about tuning the number of pseudo-categories are missing. How does the author evaluate classification accuracy with more categories than ID training datasets?
> >
> > #### A4:
> > Thank you for your question regarding the tuning of pseudo-category numbers and their impact on classification accuracy. In the unsupervised setting, we assume that the model is unaware of the number of classes in the ID dataset. As discussed in our response to Q1, the model’s ability to learn a classification logic from the ID training dataset and generalize it to the testing dataset allows us to differentiate between ID and OOD data based on performance differences.
> >
> > The choice of the number of pseudo-categories is primarily influenced by the complexity of the dataset. As shown in Table 6 of the main paper, the model achieves near-optimal OOD detection performance when the number of pseudo-categories matches the ground-truth number of categories. However, as the number of pseudo-categories increases significantly, the distinction between categories becomes less meaningful. This results in the model failing to learn a generalizable classification logic, which negatively impacts its OOD detection performance.
> >
> > To further investigate this, we analyzed the relationship between the training-testing accuracy gap (i.e., L1 distance) and the model’s ability to generalize (see Table 1). A larger gap indicates that the model has not truly learned a generalizable classification logic. Although this gap cannot directly determine the optimal number of pseudo-categories, it can serve as a useful guideline for users to select a reasonable number of pseudo-categories.
> >
> > Here is a summary of our experimental findings for different pseudo-category numbers (k) in Table 1
> >
> >
> > ### Table 1: We assessed the difference between training and testing accuracy and its effect on OOD detection performance, using the supervised benchmark setting outlined in Table 6 of our main paper. As noted, while the gap is not ideal, it can be utilized to determine the appropriate number of pseudo-categories for training classifiers on ID data.
> >
> > | **Number (K)** | **Train Acc** | **Test Acc** | **Gap** | **FPR95 ↓** | **AUROC ↑** | **Train Acc** | **Test Acc** | **Gap** | **FPR95 ↓** | **AUROC ↑** |
> > |-----------------|--------------|--------------|---------|-------------|-------------|---------------|--------------|---------|-------------|-------------|
> > |                | **CIFAR-10**                                    |                           | **CIFAR-100**                      |                          |
> > | **5**          | 99.8         | 96.7         | 3.1     | 22.18       | 94.62       | -             | -            | -       | -           | -           |
> > | **10**         | 98.5         | 95.1         | 3.4     | 24.46       | 94.01       | 94.0          | 88.7         | 5.3     | 68.23       | 74.50       |
> > | **20**         | 96.7         | 93.8         | **2.9**     | **20.23**       | **95.40**      | 92.7          | 87.1         | 5.6     | 67.15       | 76.28       |
> > | **50**         | 94.0         | 89.3         | 4.7     | 43.28       | 90.08       | 90.5          | 85.4         | 5.1     | 61.78       | 79.26       |
> > | **100**        | 91.0         | 86.7         | 4.3     | 41.56       | 87.96       | 88.9          | 84.6         | **4.3**     | 56.28       | 85.71       |
> > | **200**        | 89.1         | 80.7         | 8.4     | 60.42       | 79.02       | 83.5          | 79.0         | 4.5     | **54.92**       | **86.21**      |
> > | **500**        | -             | -            | -       | -           | -          | 8.7           | 53.2         | 5.5     | 60.72       | 78.59      |
> >
> > ---

---

> > > ### Author Response · Authors · 2024-11-21
> > > **We appreciate the reviewer’s(S5MM) insightful comments. Below, we address the questions raised by the reviewer(Part 3)**
> > >
> > > ### Questions
> > >
> > > #### Q5:
> > > Given that SSL can extract features with semantic information, is it necessary to train the model with SSL pseudo-labels?
> > >
> > > #### A5:
> > > Thank you for this valuable question. While SSL can indeed extract features with semantic information, pseudo-labels remain essential in our approach for several key reasons. Primarily, we leverage feature norm, a supervised method, for OOD sample detection, which inherently requires the model to learn to classify ID data accurately. Additionally, recent studies[3] have shown that integrating pseudo-labels into the learning process can significantly boost SSL-based feature learning. As illustrated in Table 2, we compare OOD detection results with and without the additional use of pseudo-labels to train the model. The results demonstrate that the model trained with pseudo-labels achieves superior OOD detection performance.
> > >
> > > ### Table 2: Comparison of OOD detection performance with and without pseudo-labels. In this table,”PS” stands for pseudo-labels
> > >
> > > | **Method** | **SVHN FPR ↓** | **SVHN AUR ↑** | **Textures FPR ↓** | **Textures AUR ↑** | **LSUN-C FPR ↓** | **LSUN-C AUR ↑** | **Average FPR ↓** | **Average AUR ↑** |
> > > |------------|----------------|----------------|--------------------|--------------------|------------------|------------------|-------------------|-------------------|
> > > | **SSL**    | 24.31          | 92.83          | 42.80              | 87.74              | 20.16            | 94.45            | 54.86             | 80.90             |
> > > | **SSL+PS** | **22.82**      | **93.91**      | **25.18**          | **94.25**          | **1.84**         | **99.61**        | **20.23**         | **95.40**         |

---

> > > > ### Author Response · Authors · 2024-11-21
> > > > **We appreciate the reviewer’s(S5MM) insightful comments. Below, we address the questions raised by the reviewer(Part 4)**
> > > >
> > > > ---
> > > > #### Q6:
> > > > Why do the pseudo-labels generated by SSL offer advantages over GT labels?
> > > >
> > > > #### A6:
> > > > Thank you for raising this important question. We would like to clarify that the primary advantage of our method over approaches using ground-truth (GT) labels, such as FeatureNorm, likely stems from the use of SSL pre-trained weights rather than the pseudo-labels being inherently superior to GT labels. To validate this hypothesis, we conducted an additional experiment where we fine-tuned the model pre-trained with Mixed Barlow Twins using both GT labels and pseudo-labels. The results, presented in Table 3, show that the model fine-tuned with GT labels achieves better OOD detection performance. However, the performance difference between the two approaches is relatively small, which highlights the effectiveness of pseudo-labels. It is important to note that the use of GT labels for fine-tuning requires manually annotated data, which incurs additional costs.
> > > >
> > > > ### Table 3: We compare the performance of fine-tuning the SSL model using ground-truth (GT) andpseudo (PS) labels.
> > > >
> > > > | **Method**   | **SVHN FPR ↓** | **SVHN AUR ↑** | **Textures FPR ↓** | **Textures AUR ↑** | **Average FPR ↓** | **Average AUR ↑** |
> > > > |--------------|----------------|----------------|--------------------|--------------------|-------------------|-------------------|
> > > > | **SSL+GT**   | **11.50**      | **97.86**      | **3.36**           | **99.30**          | **11.50**         | **97.74**         |
> > > > | **SSL+PS**   | 14.36          | 97.22          | 3.85               | 99.16              | 12.16             | 97.55             |
> > > >
> > > > **References:**
> > > > 1. Mark S. Graham, Walter H. L. Pinaya, Petru-Daniel Tudosiu, Parashkev Nachev, Sebastien Ourselin, and Jorge Cardoso. *Denoising diffusion models for out-of-distribution detection*. In *Proceedings of the IEEE/CVF Conference on Computer Vision and Pattern Recognition*, pp. 2947–2956, 2023.
> > > >
> > > > 2. Hamidreza Kamkari, Brendan Leigh Ross, Jesse C. Cresswell, Anthony L. Caterini, Rahul G. Krishnan, and Gabriel Loaiza-Ganem. *A geometric explanation of the likelihood OOD detection paradox*. *arXiv preprint arXiv:2403.18910*, 2024.
> > > >
> > > > 3. Wouter Van Gansbeke, Simon Vandenhende, Stamatios Georgoulis, Marc Proesmans, and Luc Van Gool. *SCAN: Learning to classify images without labels*. In *European Conference on Computer Vision*, pp. 268–285. Springer, 2020.

---

> ### Author Response · Authors · 2024-11-29
>
> Dear Reviewer S5MM,
>
> We appreciate the time and effort you have invested in reviewing our manuscript. If possible, could you kindly let us know if our responses have adequately addressed your questions? Please let us know if you have any further comments. We would be delighted to provide additional responses. Thank you.

---

### Author Response · Authors · 2024-11-25

Dear Reviewers,

We sincerely appreciate the time and effort you have dedicated to reviewing our manuscript. Your thoughtful comments and suggestions have provided invaluable guidance in improving the clarity and quality of our work.

In our rebuttal, we have thoroughly addressed all of your concerns and supplemented the paper with several new experiments to strengthen our findings. Specifically:

1. We have added detailed experiments and justifications to demonstrate why our method outperforms FeatureNorm.
2. We conducted ablation studies to explain why we selected mix-Barlow Twin over other architectures and how it contributes to the success of our approach.
3. We provided additional analysis to clarify how clustering methods and the number of pseudo-categories are chosen and their impact on model performance.
4. We included experiments with ImageNet as the in-distribution dataset to expand the scope of our evaluations and further validate our approach.

We believe these additional results and explanations address your concerns and demonstrate the robustness and generalizability of our method. If you find that our responses and new results have adequately resolved your questions, we would be grateful if you could consider revising and increasing your evaluation score.

We have made every effort to address your comments thoroughly, and we are confident that, if our paper is accepted, we will be able to revise it to its best possible form within the allocated time. Thank you again for your valuable feedback and for helping us improve our work.

Sincerely,

The authors.

---

### Meta-Review · Area_Chair_4zMw · 2024-12-19

**Metareview:**

However, while using SSL and related methods to create pseudo-labels for enhancing OOD detection is an interesting idea, this approach raises certain concerns that the authors have not adequately addressed in the current version. Additionally, I have reservations about the novelty of the proposed method. Therefore, I am inclined to recommend rejecting this paper in its present form.

**Additional Comments On Reviewer Discussion:**

As mentioned, nearly all reviewers (except one) expressed concerns about the novelty of the work. I agree with their assessment, and there are also several questions raised by the reviewers that were not addressed convincingly in the rebuttal.

---

### Decision · Program_Chairs · 2025-01-22

Reject